# K isotopes trace temporal silicate weathering intensity

Long-Fei Gou [1,2,3] ✉, He Sun[4], Hai-Ou Gu [4],
Philip A. E. Pogge von Strandmann [5], Wenshuai Li [6], David J. Wilson [7],
Jun Xiao [2], Zhi-Qi Zhao [1,8], Albert Galy [3] & Zhangdong Jin [1,2] ✉

Silicate weathering alters the biogeochemical compositions of the lithosphere, hydrosphere, and atmosphere, and thereby regulates both nutrient cycling and habitable temperatures on Earth, but tracing silicate weathering effectively remains a challenge. Potassium (K) isotopes have been proposed as a tracer of silicate weathering intensity spatially, but there is a significant gap in how and why K isotopes trace silicate weathering temporally. Here we investigate seasonal variations in dissolved K isotopes in the middle Yellow River, which drains a large area of homogeneous loess that represents the average geochemical composition of the upper continental crust, and experiences significant climatic seasonality driven by the East Asian monsoon. We find that K isotopes show strong seasonality as a function of aluminosilicate neoformation following silicate dissolution, and thus could serve as a tracer of silicate weathering intensity. We derive an empirical relationship of $\delta^{41}K_{rw} = -0.07 \times \ln(W/D) - 0.38$, where W(silicate chemical weathering)/ D(denudation) refers to silicate weathering intensity.

Silicate weathering alters the biogeochemical compositions of the lithosphere, hydrosphere, and atmosphere, and stabilizes Earth's habitability by regulating atmospheric $CO_2$ concentrations over geological timescales[1-3]. However, the factors controlling silicate weathering remain unclear, especially in deep time, with a standing debate on the relative roles of tectonic uplift[4] and climate change[3,5,6]. Such questions could be better addressed through the development of new tracers of silicate weathering that can be applied in both the modern day and the geological past.

Potassium (K) is almost exclusively hosted in silicates[7-9] and has two stable isotopes, $^{39}K$ and $^{41}K$, which are proposed to fractionate during dissolution, adsorption, and incorporation[7,10,11]. As such, K isotopes ($\delta^{41}K$) are a promising tracer of chemical weathering intensity

(the ratio of chemical weathering and total denudation flux)[7-12]. For example, a weak relationship between the annually-averaged chemical weathering intensity and $\delta^{41}K$ values in river waters ($\delta^{41}K_{rw}$) was reported spatially at a global scale[8]. However, seasonal variations in $\delta^{41}K_{rw}$ values have not yet been demonstrated, despite the reported sensitivity of silicate weathering to climatic parameters[13-15]. If $\delta^{41}K_{rw}$ values do serve as a tracer for spatial variations in chemical weathering intensity, then we could expect them to also respond due to silicate weathering to climate forcing[7-9,16].

Here, we show how $\delta^{41}K_{rw}$ can act as a tracer for temporal variations in silicate weathering intensity under variable climatic conditions, including pronounced temperature variations and extreme hydrological events. To this end, we determine $\delta^{41}K_{rw}$ values in a (semi-

[1]State Key Laboratory of Loess Science, Department of Geography, Chang'an University, 710054 Xi'an, China. [2]State Key Laboratory of Loess Science, Institute of Earth Environment, Chinese Academy of Sciences, 710061 Xi'an, China. [3]Centre de Recherches Pétrographiques et Géochimiques, UMR7358, CNRS, Université de Lorraine, 54500 Vandoeuvre les Nancy, France. [4]School of Resources and Environmental Engineering, Hefei University of Technology, 230009 Hefei, China. [5]Mainz Isotope and Geochemistry Centre (MIGHTY), Institute of Geosciences, Johannes Gutenberg-Universität Mainz, 55128 Mainz, Germany. [6]School of Earth Sciences, China University of Geosciences, 430074 Wuhan, China. [7]London Geochemistry and Isotope Centre (LOGIC), Department of Earth Sciences, University College London, Gower Street, WC1E6BT London, UK. [8]School of Earth Science and Resources, Chang'an University, 710054 Xi'an, China. ✉e-mail: lfgou@chd.edu.cn; zhdjin@ieecas.cn

)arid region with limited vegetation using high-resolution temporal sampling in the middle reaches of the Yellow River (Fig. S1). In arid environments, low vegetation cover minimizes biological uptake and thus reduces the biological effects on K isotopic fractionation. This river system drains across easily erodible and relatively homogeneous loess (Fig. S1), which closely represents the average chemical composition of the upper continental crust (UCC[17]). Pronounced seasonal climate changes driven by the East Asian monsoon make the Yellow River a highly suitable setting to define the seasonal response of $\delta^{41}K_{rw}$ to climate. We find significant $\delta^{41}K_{rw}$ seasonality as a function of aluminosilicate neoformation after silicate dissolution, suggesting that it can serve as a tracer of silicate weathering intensity. We also establish an empirical relationship of $\delta^{41}K_{rw} = -0.07\ \ln(W/D) - 0.38$, where W(silicate chemical weathering)/D(denudation) refers to silicate weathering intensity.

## Results

### Seasonal variations in temperature, discharge, and physical erosion rates

During the sampling year of 2013, the water temperature continuously increased from a January minimum of 0 °C to an August maximum of 29 °C, and then gradually decreased (Fig. 1). The water discharge at the Longmen hydrological station was 24.5 km³/yr in 2013. During the dry season, there were low values of water discharge in January–February, which then peaked in March, and then declined to a minimum of 152 m³/s in May (Fig. 1). We defined the first small water discharge peak as an "ice melting interval" because it was a result of ground snow melting from 16$^{th}$ March to 13$^{th}$ April when the air temperature was above 0 °C. During the monsoon season (June to mid-September), the consistently high-water discharge (> 600 m³/s; Fig. 1) reflected the frequent, monsoon-driven precipitation within the Yellow River basin. Notably, there was a storm event from 22$^{nd}$ to 25$^{th}$ July, which resulted in a maximum water discharge of 2400 m³/s[18-21]. After the monsoon season, the water discharge decreased gradually to relatively low values for October to December. All the waters of the middle Yellow River were alkaline with pH values between 7.05 and 8.71[21].

The Yellow River is highly sediment-laden, contributing ~10% of the global sediment input to oceans[22]. Seasonal variations in the suspended particulate matter (SPM) flux in the middle Yellow River span almost five orders of magnitude (Fig. 1). The SPM flux was low and fairly constant during the dry season, with a spike during the ice melting interval, whereas high concentrations and flux of SPM characterized the monsoon season (Fig. 1). The highest concentrations and flux of SPM were recorded during the storm event in July. Overall, physical erosion rates (PER) during the monsoon season were one to four orders of magnitude higher than those during the dry season (Figs. 1, S2), suggesting that abundant loess was eroded into the river during the monsoon season[18-21].

### K concentrations

The time series of the K$^+$ concentrations ([K$^+$]) and $\delta^{41}K_{rw}$ values of the Yellow River water are shown in Fig. 1, Table S1. The mean [K$^+$] in the river waters was 110 μmol/L, ranging from 89 μmol/L for the storm event to 163 μmol/L during the winter, with significant seasonal variations. These [K$^+$] values fit within the range of global large rivers (7 to 180 μmol/L[7,8]) and the Mun River in Thailand (58 to 360 μmol/L[16]). There was no correlation of [K$^+$] with [SO$_4^{2-}$], [Cl$^-$], or [NO$_3^-$]. Similar to observations from global rivers[5], the riverine K$^+$ flux was positively correlated with the PER in the middle Yellow River (Fig. S2).

Sequential extraction results for K from the Lingtai loess are given in Table S3, aiming to extract the salt ('evap', water-soluble fraction), carbonate ('carb', 5% acetic acid-soluble fraction), and the silicate ('sil', residue after sequential extraction) fractions of the loess[23,24]. Generally, both the salt and carbonate fractions contain relatively little K, with 0.14 ± 0.12 mg/g and 0.62 ± 0.24 mg/g, respectively[23,24]. In contrast, the silicate fraction contains K concentrations two orders of magnitude higher than those in the salts and carbonate fractions, with a mean of 18.3 ± 4.3 mg/g that is similar to the composition of the UCC (19.00 ± 2.99 mg/g[11,17]). Given that the SPM in the middle Yellow River has the same chemical, mineralogical, and Li, Mg, and Ba isotopic compositions as the loess[18-20,25], we used 18.3 ± 4.3 mg/g of the loess as [K]$_{SPM}$[11,17,23].

The mean [K$^+$] of the rainwater samples was ~30 μmol/L (a range of 17–47 μmol/L, Table S2), while the mean rainwater K/Cl ratio of 0.33 excludes a recycled sea-salt origin, since sea-salt has a typical K/Cl ratio of 0.02 and $\delta^{41}K$ values of 0.12 ± 0.07‰[8,26]. Therefore, the high [K$^+$] in the rainwater was likely related to the high dust contributions in the Asian interior[27], as similarly inferred for Li, Mg, Sr, and Ba isotopes[18-20,27]. A sewage water sample collected in farmland had [K$^+$] of 827 μmol/L (Table S2), which is higher than any other samples collected in the middle Yellow River. A groundwater sample had [K$^+$] of 62 μmol/L (Table S2), which is higher than the rainwater but slightly lower than the river waters.

### K isotopes

Clear seasonality was observed in the $\delta^{41}K_{rw}$ values of the middle Yellow River water, which ranged between −0.37‰ and +0.27‰, far beyond the typical analytical 2 s.d. of 0.11‰ (Fig. 1). This finding represents the first reported example of seasonal $\delta^{41}K_{rw}$ variations, which span the overall range of $\delta^{41}K_{rw}$ variations observed globally (0.65‰), even taking spatial variations into consideration[7,8,16] (Fig. S3). Generally, during the dry and cold seasons, $\delta^{41}K_{rw}$ values were low (−0.37‰ to −0.10‰). In contrast, during the warm and wet monsoonal season, $\delta^{41}K_{rw}$ values were high (−0.10‰ to +0.27‰; Fig. 1). A similar pattern (though smaller magnitude) was also observed in the Yangtze River, with the wet season corresponding to high $\delta^{41}K_{rw}$ values and the dry season to low $\delta^{41}K_{rw}$ values[7].

Sequential extraction experiments on five loess samples gave $\delta^{41}K_{evap} = +0.03 ± 0.30‰$, $\delta^{41}K_{carb} = -0.17 ± 0.08‰$, and $\delta^{41}K_{sil} = -0.36 ± 0.12‰$ (Table S3)[23,24]. The $\delta^{41}K_{sil}$ values are similar to the bulk silicate earth (BSE) and the UCC ($\delta^{41}K$ of −0.48‰ to −0.35‰[11,28]; Fig. S3). Heavy K isotopes are preferentially incorporated into K-bearing evaporites due to the equilibrium isotope effects that result from changes in coordination number, bond length, and bond strength[29,30]. Hence, we suggest that the evaporites and carbonates in loess are mainly secondary minerals that formed after dissolution of the primary eolian loess, because higher $\delta^{41}K$ values are observed for the secondary evaporites and carbonates than for the silicates in the loess, as expected[30].

The rainwater sample had a very negative $\delta^{41}K$ value of −0.68 ± 0.13‰ (Table S2), together with K/Cl molar ratios between 0.15 and 0.48, further supporting that it was not of sea-salt origin. Given its very different composition from the $\delta^{41}K_{rw}$ and the occurrence of higher $\delta^{41}K_{rw}$ values at times of high-water discharge (Fig. 1), rainwater input is likely to be negligible. Similarly, a sewage sample collected on farmland had a $\delta^{41}K$ value of −0.50 ± 0.03‰ (Table S2), which also excludes a significant anthropogenic K$^+$ source to the middle Yellow River. A groundwater sample had a $\delta^{41}K$ value of −0.05 ± 0.00‰, which is comparable to the annual-average $\delta^{41}K_{rw}$ values in the middle Yellow River.

## Discussion

The export of K as solids in suspension (K$_{SPM}$) dominates the overall K flux in the middle Yellow River, averaging ~60% throughout time (Fig. S4). The highest proportion of K transport via solids occurs in the monsoon season, at typically ~95% and peaking at 99.7%. During the ice-melting interval, the proportion of K transported as solids also increases to ~95% (Fig. S4). The temporal patterns in the proportion of K exported as solids and in the total SPM concentration are similar to the pattern of seasonal variations in the $\delta^{41}K_{rw}$ values (Fig. S5),

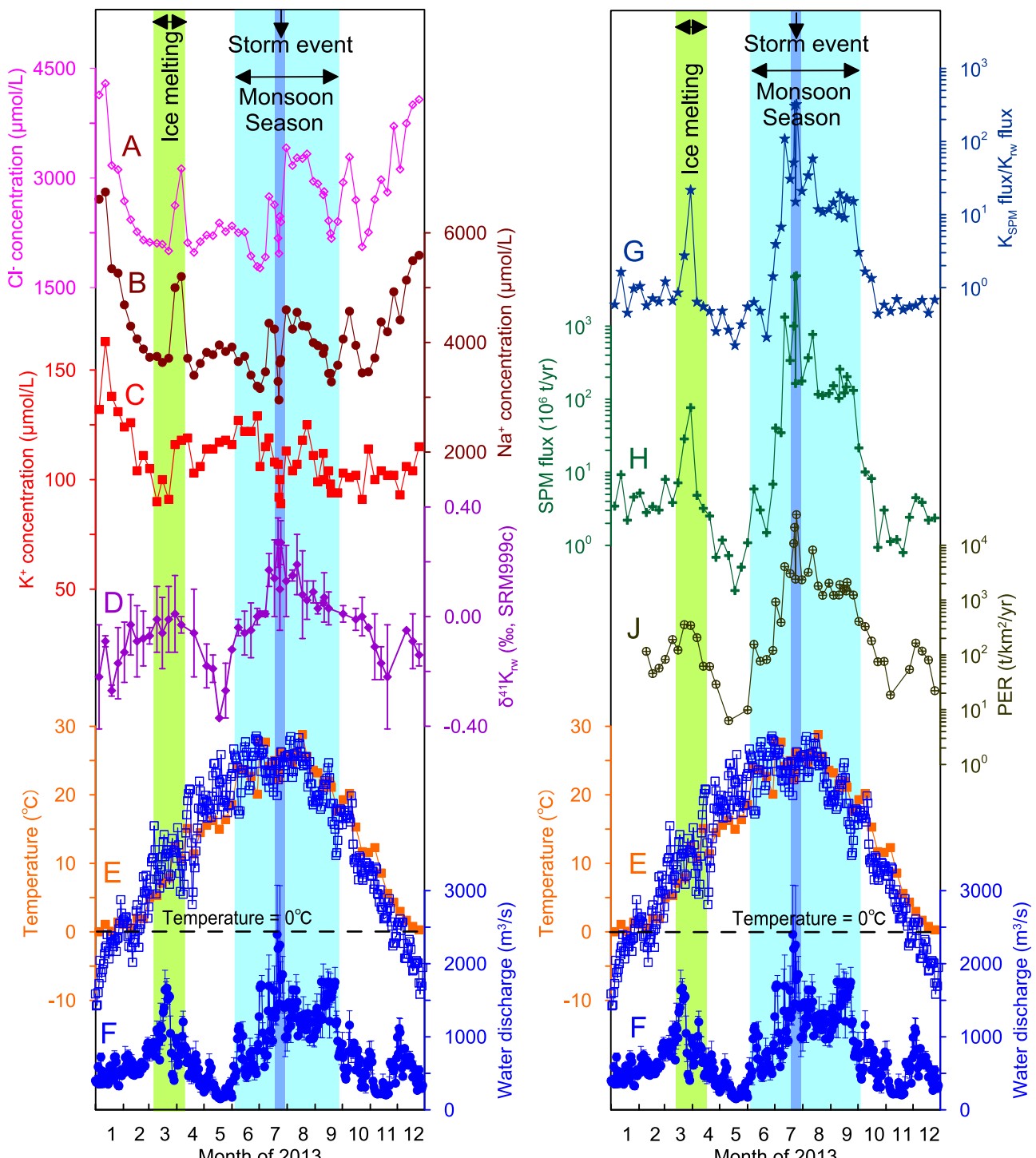

**Fig. 1 | Seasonal hydrological and geochemical parameters for the middle Yellow River during 2013.** **A** Cl⁻ concentration ([Cl⁻]), **B** Na⁺ concentration ([Na⁺]), **C** K⁺ concentration ([K⁺]), and **D** $\delta^{41}K_{rw}$ values of river water collected weekly at the Longmen hydrological station. **G** Ratios between SPM K flux and dissolved K⁺ flux, **H** suspended particulate matter (SPM) flux, and **J** physical erosion rate (PER, from Zhang et al., 2015)[21] at the Longmen station. Also shown for comparison are (**E**) water temperatures (orange squares) and air temperatures (open blue squares), and (**F**) water discharge. The ice-melting interval (16th March to 13th April), monsoon season (June to mid-September), and a storm event (22nd to 25th July) are shaded green, pale blue, and dark blue, respectively.

suggesting that the K isotopic behavior is closely related to the SPM content, via adsorption and/or incorporation processes.

Mass-balance calculations (see Supplementary Information) show that the weighted average silicate dissolution dominates the riverine K⁺ budget (73.3 ± 6.3%), while evaporite dissolution contributes limited K⁺ (25.8 ± 6.3%) to the middle Yellow River (Fig. 2). In contrast, although the K⁺ contents of the carbonate leachates of the loess seem to be

higher than expected for a pure carbonate and may inevitably also contain a non-carbonate K signal (Table S3), carbonate dissolution (0.06 ± 0.02% as an upper limit), atmospheric input (0.90 ± 0.10%), and anthropogenic input (0.03 ± 0.01%) play a negligible role in the riverine K⁺ budget in the middle Yellow River (Fig. 2). These findings for the elemental budget are supported by the large difference between the $\delta^{41}K_{rw}$ values and the $\delta^{41}K$ values of both rain and anthropogenic inputs

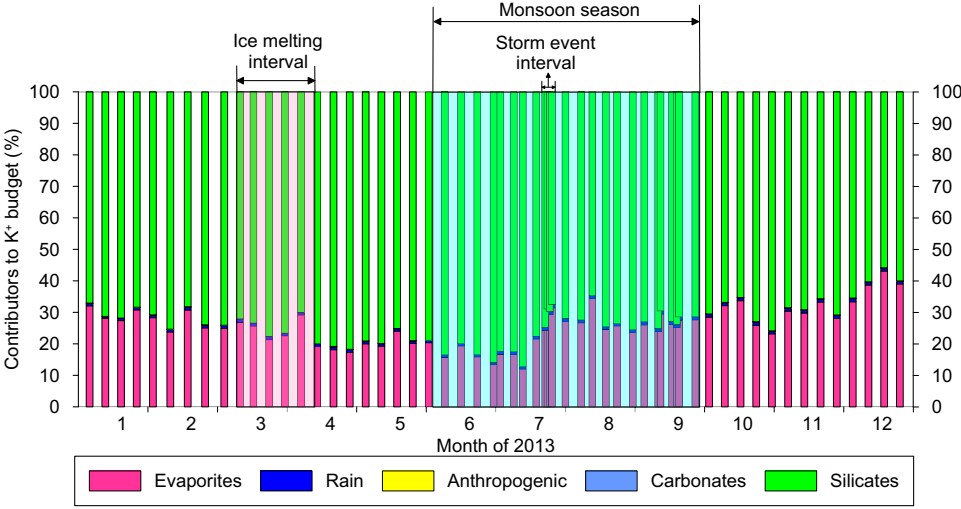

**Fig. 2 | Partitioning of the dissolved K⁺ budget into five end-members, i.e. evaporites, rain, anthropogenic, carbonates, and silicates.** The weathering of silicates dominates the dissolved K⁺ budget (an annual average of 73.3 ± 6.3%), while another significant contributor is evaporites (25.8 ± 6.3%), whereas the other contributions are negligible. See supplementary text for the calculations.

(Table S2). The 25.8 ± 6.3% input of K⁺ from evaporites is comparable to its contribution to the dissolved Li⁺ budget in this river[20]. The lack of a relationship between the proportions of K⁺ from any sources (i.e. silicates, carbonates, and evaporites) and $\delta^{41}K_{rw}$ values rules out a dominant control on the $\delta^{41}K_{rw}$ variability by mixing between those sources (Fig. S6). Although atmospheric K⁺ inputs (e.g. biomass burning, traffic emissions) are not significant in this basin, they may be relevant in other systems with higher atmospheric deposition, which would thus merit further investigation. Overall, silicate dissolution dominates the riverine K⁺ budget, and the riverine K⁺ flux is positively correlated with the PER (Fig. S2), both of which represent the preconditions for using K isotopes as a tracer for silicate weathering[8,16].

Fertilizers are excluded as a contributor to the K⁺ budget of the middle Yellow River, due to the sparse farmland and very negative $\delta^{41}K$ values of a sample from farmland (Table S2). Plant uptake can favor both light or heavy K isotopes[31], but we exclude the possibility of a vegetation control on $\delta^{41}K_{rw}$ in the middle Yellow River for three reasons. First, vegetation is very sparse in the (semi-)arid middle Yellow River[32]. Second, plant growth is enhanced after the ice-melting period, but the most negative $\delta^{41}K_{rw}$ values occur at this time, while there are similar $\delta^{41}K_{rw}$ values during both the ice-melting interval and the monsoon season (Fig. 1). Third, plant defoliation should contribute a large amount of K⁺ into the basin, but the K/Sr ratio smoothly decreased after August (Fig. S7). In contrast, the $\delta^{41}K_{rw}$ values are positively correlated with the K⁺ flux and the chemical weathering rate (Fig. S8), suggesting a silicate weathering control on the K⁺ budget, because evaporite-sourced K⁺ should only be sensitive to water discharge rather than to chemical weathering rate. Together with the absence of source mixing relationships (Fig. S6) and the negligible carbonate-sourced K⁺ (Fig. 2), the $\delta^{41}K_{rw}$ should predominantly reflect natural weathering processes, i.e. K⁺ release from silicates and K⁺ uptake by SPM[7,8,16] (Fig. S5).

The initial dissolution of K from rocks could kinetically release light K isotopes into the fluid, while the fractionation factor (α) during dissolution seems to be insensitive to mineralogy[33]. However, K isotopes have been shown to reach equilibrium after ~10 h in laboratory experiments[33], whereas the interaction timescale between fluids and rocks in large watersheds ranges from seconds to years and is likely often in a disequilibrium state[34]. Therefore, K isotope fractionation during dissolution should be considered as a possibility. Here, we employed both Rayleigh and batch models to simulate the dissolution processes for short and long timescales (Fig. 3). Such modeling only

considers thermodynamic equilibrium and not any kinetic processes potentially affecting K partitioning between solid and aqueous phases.

The combined dissolution and incorporation process is modeled by assuming a constant α between fluid and SPM during each of the dissolution and subsequent incorporation processes (Fig. 3). The $\alpha_{SPM-fluid}$ for dissolution is obtained from published dissolution experiments, ranging between 1.00045 and 1.00105[35]. We further expand the range from 1.00000 to 1.00105[35] to cover a wider set of possibilities, because the above experiment was carried out in acidic conditions that may not be representative of the Yellow River[35]. The Rayleigh fractionation equation can be written as $\delta^{41}K_{rw} = (\delta^{41}K_{loess} + 1000)f^{(\alpha-1)} - 1000$, where $\delta^{41}K_{loess}$ is the K isotopic compositions of loess, and $f$ is the fraction of K remaining in the river water normalized to Na*, calculated from $[K/Na^*]_{rw}/[K/Na]_{loess}$[36] (here Na* = [Na⁺] − [Cl⁻] to eliminate the impact of evaporite-sourced Na⁺; Fig. 3). The α value for dissolution seems to be insensitive to mineralogy[34], but it varies during incorporation due to the variable site-preference of K⁺. However, there is no α value available from silicate synthesis experiments so far, so we used $\alpha_{SPM-fluid} = (^{41}K/^{39}K)_{clay}/(^{41}K/^{39}K)_{rw}$ in a range between 0.99955 and 0.99895 obtained from our data (Table S1 for water and Fig. S3 for clays). These α values are broadly in the range of the 0.99976 deduced from ab initio calculations for equilibrium fractionation between fluids and illite[35] and 0.99937 and 0.99800–1.00000 estimated from various natural observations[12,37].

Although the above selection of α values involves some uncertainties, the modeled trends cover our observations regardless of the exact choice of α values (Fig. 3). Simple incorporation of K⁺ seems unable to explain the dataset, both because a source for dissolved K⁺ is required and because the theoretically-calculated K/Na* ratios would be too high (for a given $\delta^{41}K_{rw}$ value) compared to the observed values (Fig. 3). However, mixing between the signatures of Rayleigh and/or batch dissolution, which release light K isotope, and incorporation, which fractionates river water towards heavy K isotope, could then explain the observations (Fig. 3).

A control of mass-dependent diffusion across the rock–fluid interface on K isotope variations in the Yellow River can be ruled out directly. For K⁺ in water, the diffusion coefficient follows $D \propto m^{-\beta}$, with $0 \le \beta < 0.20$[38], where D, m, and β refer to the diffusion coefficient, the mass of the diffusing particle, and the mass-scaling exponent, respectively. This relationship means that heavier or lighter ions would

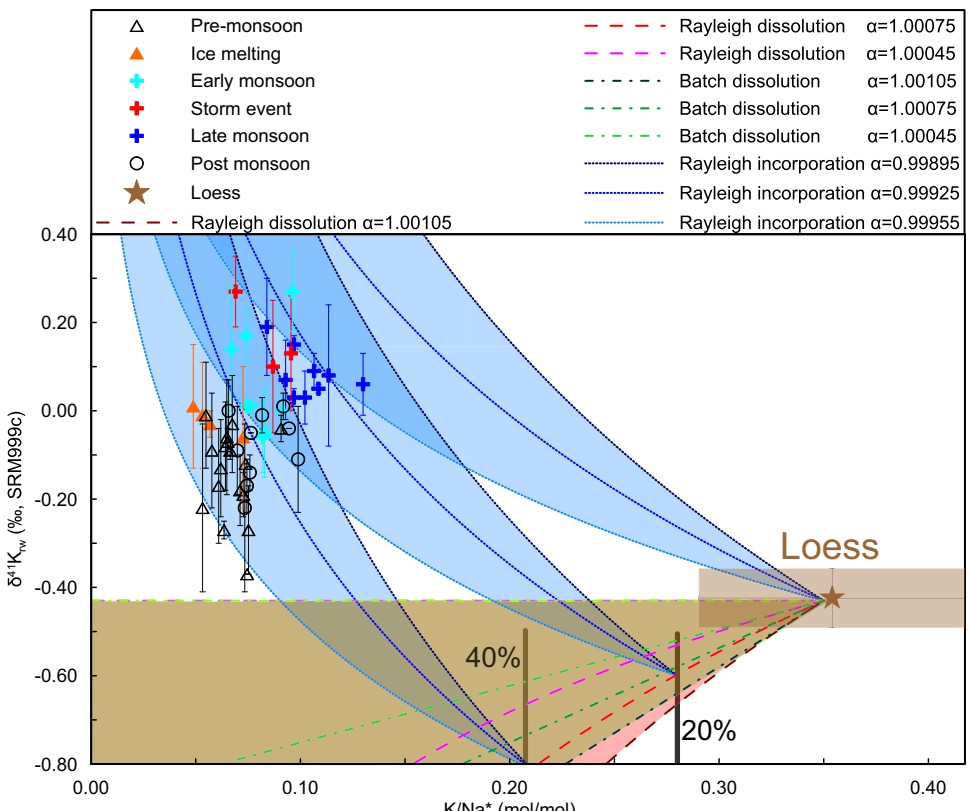

**Fig. 3 | Potassium isotope fractionation during dissolution and aluminosilicate neoformation.** $\delta^{41}K_{rw}$ versus K/Na* ratios, where Na* = [Na⁺] – [Cl⁻]. The curves indicate modeled silicate dissolution (batch or Rayleigh fractionation) followed by aluminosilicate neoformation (Rayleigh fractionation), with potential fractionation factors based on Li et al. (2021)[33]. The loess K/Na* ratio is calculated from Sauzeat et al. (2015) and Huang et al. (2020)[11,36]. The labeled bars show the proportion of remaining K⁺ relative to the conservative Na⁺. The pink and brown shadings show the feasible zones from Rayleigh and batch dissolution, respectively.

diffuse at slightly different rates. At the molecular level, this feature should affect how long water molecules stay in the first solvation shell around dissolved ions. If diffusion were important, then during the dry season (i.e. when river mixing is weaker), we would expect stronger isotope fractionation, leading to heavier Li and K isotopes in the water. However, the opposite trend is observed for K isotopes (Fig. 3). In addition, we observed a thermodynamic temperature control on seasonal variations in Li isotopes in the Yellow River[20], and since there is no correlation between Li and K isotopes, we rule out any dominant temperature effect on the K isotopes (Fig. S9).

Since the $\delta^{41}K_{rw}$ values are positively correlated with SPM concentrations (Fig. S5), and SPM mainly derives from erosion and aluminosilicate neoformation (Fig. S2), we also consider the potential for K isotope fractionation due to incorporation and/or adsorption processes after K release into fluids. However, we exclude adsorption as a main factor for two reasons. First, by analogy with evidence from nuclear magnetic resonance (NMR) spectroscopy on Li behavior, outer-sphere K is suggested to be fully hydrated and less isotopically fractionated relative to the source fluid[39], whereas experiments show that K adsorption preferentially removes heavy K isotopes onto surficial minerals[40]. However, we observe high $\delta^{41}K_{rw}$ at times with high SPM concentrations (Fig. S5), implying the removal of light K isotopes, which could not be explained by such adsorption processes. Second, Ba isotopes suggest Ba²⁺ removal via adsorption, whereas K and Ba isotopes show no relationship (Fig. S10). In contrast, incorporation into clays favors light K⁺, although interlayer K could get fully hydrated and be less fractionated relative to the source fluid[9], which is supported by ab initio calculations[35] and natural observations[12,37].

Overall, the processes of silicate dissolution followed by incorporation into clays appear to dominate the $\delta^{41}K_{rw}$ variability. A control on

$\delta^{41}K_{rw}$ by clay formation is also supported by the co-variation between dissolved K/Na* and Si/Na* ratios (Fig. S11), which could be explained by simultaneous removal of Si and K⁺ during aluminosilicate neoformation. Furthermore, we calculated the saturation indices (SI) of various minerals in the sampled waters using PHREEQC (version 3; Table S4)[41], considering parameters including pH, water temperature, Ca²⁺, K⁺, Mg²⁺, Na⁺, F⁻, Cl⁻, NO₃⁻, SO₄²⁻, CO₃²⁻, Si, Sr²⁺, Ba²⁺, Al, Fe, and Mn concentrations. Saturation indices > 0 calculated by PHREEQC for some K-bearing aluminosilicates (Table S4)[41], together with the reported illite neoformation in microenvironments (despite overall undersaturation)[42], support that clay formation could be the main driver of $\delta^{41}K_{rw}$.

The SPM in rivers mainly results from physical erosion and aluminosilicate neoformation[5], whereas dissolved K⁺ in rivers is derived from silicate dissolution (Fig. 2). Riverine $\delta^{41}K_{rw}$ values are dominated by the isotopic fractionation during incorporation following dissolution (Fig. 3). Therefore, a high ratio between the dissolved K⁺ flux and the solid K flux transported via SPM reflects a high silicate weathering intensity (i.e. most K is dissolved), and corresponds to low $\delta^{41}K_{rw}$ values due to solid dissolution (Figs. 4, 5). In contrast, a low ratio between the dissolved K⁺ flux and the solid K flux transported via SPM reflects a low silicate weathering intensity (i.e. most K is in solid), and corresponds to high $\delta^{41}K_{rw}$ values due to light K⁺ removal into clays (Figs. 4, 5). A control through this process of aluminosilicate neoformation is also supported by a broad negative co-variation of $\delta^{41}K_{rw}$ values with $\delta^{26}Mg_{rw}$ data (Fig. S12)[18]. Although K⁺ is mainly sourced from silicate dissolution, there is a non-negligible evaporite input in the middle Yellow River, so that we use W/D to reflect silicate weathering intensity, where W is the silicate chemical weathering flux and D is the total denudation[21] (Fig. 5). As such, $\delta^{41}K_{rw}$ values are expected to negatively correlate with W/D changes through time[7], as observed,

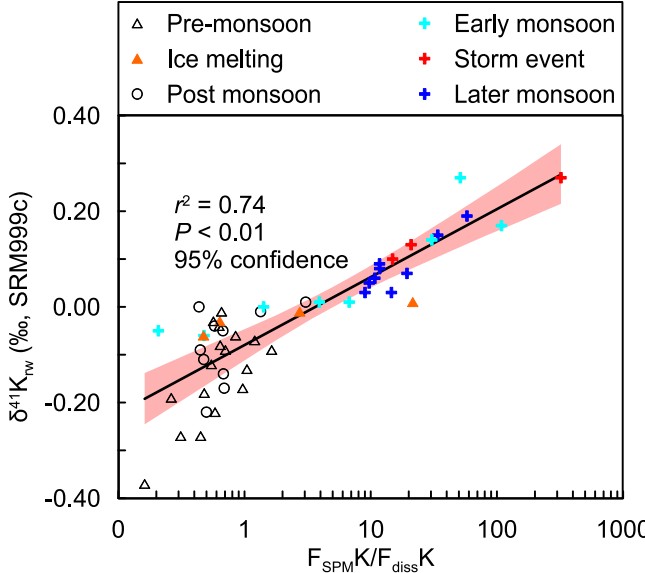

**Fig. 4 | Particulate control on riverine K isotope variations.** Correlation of $\delta^{41}K_{rw}$ values with the ratio of thesolid K flux to the dissolved K flux (on a logarithmic scale).

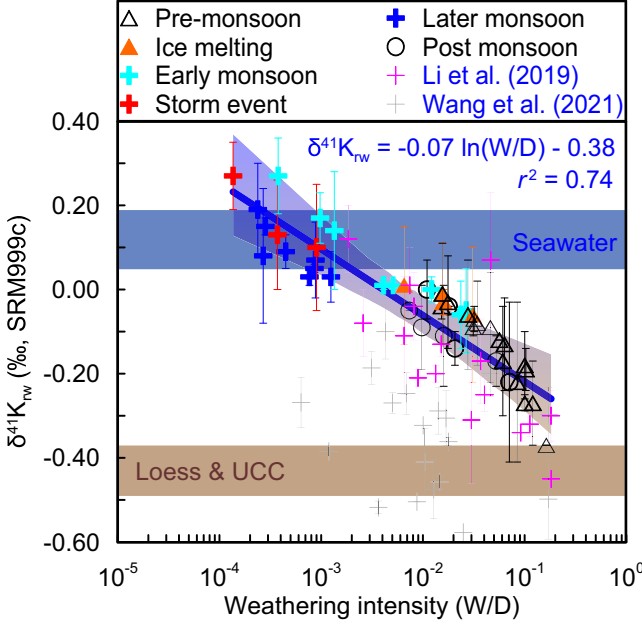

**Fig. 5 | Potassium isotopes as a tracer of silicate weathering intensity.** Cross plot of spatial and temporal variations in $\delta^{41}K_{rw}$ versus silicate weathering intensity (W/D, where W = silicate weathering rate, D = denudation rate). Data are from the Yellow River (this study, 51 samples), Li et al. (2019)[7] for the Yangtze and other rivers, and Wang et al. (2021)[8] for global large rivers. The blue line is a regression between $\delta^{41}K_{rw}$ and W/D based on data for the Yellow and Yangtze rivers. The $\delta^{41}K_{rw}$ data from global large rivers are excluded from the regression because they represent snapshot sampling, which may not reflect the inter-annual average W/D conditions. Values for the $\delta^{41}K$ of seawater are from Wang et al. (2021)[8], and the upper continental crust (UCC) and loess data are from Huang et al. (2020)[11].

from which we derive an empirical correlation of $\delta^{41}K_{rw} = -0.07 \times \ln(W/D) - 0.38$ (Fig. 5). Unlike riverine Li isotopes which are also proposed to reflect silicate weathering intensity due to fractionation during incorporation into secondary minerals, but with a "boomerang" pattern[43], K isotopes show a unidirectional pattern with W/D (Fig. 5), which may be

beneficial in facilitating the application of $\delta^{41}K_{rw}$ as a tracer of silicate weathering intensity.

Considering that $\delta^{41}K_{rw}$ values reflect an instantaneous snapshot between silicate dissolution and aluminosilicate neoformation, with strong seasonality, the longer-term (e.g. annually-averaged) W/D may not be expected to co-vary with instantaneous spot-sampled $\delta^{41}K_{rw}$ values in a spatial sampling strategy[8]. Nevertheless, it is interesting to note that the $\delta^{41}K_{rw}$ data reported from global rivers seem overall lower than our observed values, and even lower than the UCC for a few samples (Fig. 5), which requires further investigation. However, we suggest this discrepancy could be attributable to three factors: (1) some of the spatial samples were filtered with a 0.45 μm filter that could potentially contain more colloidal material comprising neo-formed aluminosilicates with light incorporated K isotopes[8] (see clay values in Fig. S3); and/or (2) unacidified samples may have been susceptible to contamination by biota; and/or (3) the calculated weathering intensity (W/D) based on sampling several decades ago could have changed significantly in recent years (e.g. the Yellow River has only a 20% SPM yield today compared to half a century ago[22,44]). In combination, we contend that the relationship between W/D ratios and $\delta^{41}K_{rw}$ values has typically been obscured in spatial investigations[8,16], whereas the high-resolution time series sampling strategy used here, and which captured a once in a century storm event, demonstrates that $\delta^{41}K_{rw}$ values negatively correlate with W/D ratios. Hence, $\delta^{41}K_{rw}$ values provide a novel tool for assessing silicate weathering intensity. However, we would encourage further research on a wider range of modern river systems to better validate this empirical relationship and to reveal any environmental circumstances in which it might be significantly altered or break down.

Global seawater ($\delta^{41}K_{sw}$ ~ + 0.12‰)[45] is significantly isotopically heavier than the UCC ($\delta^{41}K$ ~ −0.44‰)[11], which has mainly been attributed to $K^+$ removal through sediment sinks, early diagenesis, oceanic crust alteration, and reverse weathering[12,45–48]. In contrast, $K^+$ release from mid-ocean ridge vent systems is limited and also has low $\delta^{41}K$ values (−0.46‰ or −0.15‰)[37]. The reported average terrestrial weathering input of K isotopes ($\delta^{41}K = -0.38 \pm 0.04$‰)[8] is also low, but constraints on seasonal variability have been lacking until now[7,8]. Here we show that $\delta^{41}K_{rw}$ could vary significantly on seasonal timescales and can reach values as high as +0.27‰ under extreme incongruent weathering conditions (W/D < 0.0001, Fig. 5) in which large amounts of nucleation help to drive aluminosilicate neoformation. Our findings indicate that major temporal $\delta^{41}K_{rw}$ variations in riverine inputs, possibly arising from Tibetan Plateau uplift and other orogenic events during the Cenozoic, could potentially explain the $\delta^{41}K_{sw}$ evolution without any other processes[12,45–48].

In deep time, the weathering of the UCC can be conceptualized as a globally integrated source of dissolved $K^+$ to the oceans. As such, the variability in marine K isotopic compositions preserved in sedimentary archives may reflect silicate weathering intensity on Earth through time. Since carbonates are vulnerable to biological fractionation of K isotopes[49], oceanic authigenic clay minerals (e.g. illite, glauconite, and Fe-smectite) with a more constant (albeit likely temperature-dependent) fractionation factor from seawater could potentially serve as a robust archive of paleo-seawater K isotopes. Such records could enable the effective reconstruction of long-term changes in Earth's weathering-climate feedback[1–3].

## Methods

Information on the field sampling, extraction experiment on the loess, geochemical analyses, and K isotope analyses is described below.

### Field sampling

A total of 60 river water samples were collected weekly in 2013 at the Longmen hydrological station (35°40′06.43″ N, 110°35′22.88″ E; Table S1). This station is located in the middle reaches of the Yellow

River, after the convergence of most tributaries draining the Chinese Loess Plateau (Fig. S1). Note that four river water samples (LM13−31 to 13−34) were collected daily during a storm event in July[18–21]. Three rainwater samples were collected in July and August 2013 at the station to assess atmospheric inputs, and a sewage sample (TKT1) and a groundwater sample (T10GW) were collected in farmland adjacent to the station to constrain the composition of anthropogenic and groundwater K inputs[18–21] (Table S2).

All river water samples were collected 0.5 m below the river surface in the central part of the river channel. For each sample, water temperature, pH, electrical conductivity (EC), and total dissolved solids (TDS) were measured in situ. All water samples were filtered through 0.2 μm nylon filters on site. Filtered water samples were stored in pre-cleaned polyethylene bottles, acidified to pH <2 with distilled $HNO_3$, and stored at 4 °C, before analysis of major cationic concentrations and K isotopes.

### Sequential extraction experiment for loess
Five fresh loess samples were collected from five typical layers of the loess profile at Lingtai and were subjected to sequential extraction for K isotopes (Table S3). Briefly, 0.5 g of milled loess was leached with 18.2 MΩ.cm water for 5 min, and centrifuged and filtered via manual filters to collect the water-soluble fraction[21]. The residue was then leached for 2 h with 5% acetic acid (HAc) at 75 °C, and then centrifuged to collect the carbonate fraction[23,24]. The residues of the leaching procedure were digested with HF−HCl−$HNO_3$ to constrain the silicate fraction.

### Geochemical analyses
The concentrations of major ions for all samples were reported by Zhang et al. (2015)[21]. Major cations (including $K^+$) were analyzed by a Leeman Labs Profile inductively coupled plasma atomic emission spectroscopy (ICP-AES), with a relative standard deviation (RSD) better than 5% according to in-house standards and reference materials. Major anions ($F^-$, $Cl^-$, and $SO_4^{2-}$) were measured by ion chromatography (ICS 1200), and $NO_3^-$ was measured by a Skalar continuous flow analyzer, with an RSD better than 5%. Alkalinity (expressed as $HCO_3^-$) was measured by a Shimadzu Corporation total organic carbon analyzer (TOC-$V_{CPH}$), with an RSD better than 5%. The percent charge balance error (CBE), as a measure of the data quality, is given by the equation $[CBE (\%) = (TZ^+ - TZ^-)/(TZ^+ + TZ^-) \times 100]$, where $TZ^+ = 2Ca^{2+} + 2Mg^{2+} + K^+ + Na^+$, $TZ^- = Cl^- + 2SO_4^{2-} + NO_3^- + HCO_3^-$, with an average better than ± 5%.

### K isotope analyses
Pre-treatment and analyses of the K isotopic compositions of all samples of river water, rainwater, sewage water, and groundwater were performed in an ultraclean room (class 1000) at the Hefei University of Technology (HFUT)[50]. Typically, 2 mL of river water, sewage water, and groundwater, and ~20 mL of rainwater were used, enabling 1 μg $K^+$ to be retrieved. These samples were dried down after organic matter digestion (using 1 mL of concentrated $H_2O_2$ and $HNO_3$), and then re-dissolved in 0.5 M $HNO_3$, before K purification by column chromatography. The samples were passed twice through Savillex® PFA micro-columns (0.64 cm × 8 cm, inner diameter and length, respectively) filled with 2 mL resin (Bio-Rad® AG50W X-8, 200-400 mesh) for cation exchange chromatography, with 0.5 M $HNO_3$ as eluent[50]. The columns were pre-cleaned with 12 mL of an acid mixture of 6 M $HNO_3$ + 0.5 M HF. The purified K fraction was re-dissolved in 2% $HNO_3$ and diluted in order to obtain 200 μg/L of K for K isotope measurements. The total procedural blank of this method was less than 10 ng K, which is negligible relative to 1 μg of K analyzed in each sample[50].

Isotopic analyses were conducted on a *Neptune Plus* multi-collector inductively coupled plasma mass spectrometer (MC-ICP-MS, Thermo Fisher, Germany) at the HFUT. Analyses used a "Continuous-Acquisition-Method" and sample-standard-bracketing (SSB) with the international standard NIST SRM999c for instrumental mass fractionation correction[50]. The K isotopic composition ($δ^{41}K$) is reported using the delta-notation in per mil:

$$δ^{41}K(‰) = \left( \frac{\frac{41_K}{39_K}(\text{sample})}{\frac{41_K}{39_K}(\text{SRM999c})} - 1 \right) \times 1000‰ \qquad (1)$$

where SRM999c is the average value of the standard solution measured immediately before and after each sample. Note that some previous data were reported relative to different standards, i.e. SRM3141a, SRM918b, and SRM193[51–53]. All standards were demonstrated to be indistinguishable for their $δ^{41}K$, within current analytical precision[50]. The $δ^{41}K$ value was obtained from triplicate measurements, from which mean values and the standard deviation (2 s.d.) were calculated for each sample.

In order to validate the measured K isotope data, four in-house standards (GBW−K, GSB−K, QC−K, and ST−K) were analyzed repeatedly and yielded $δ^{41}K$ values of 0.29 ± 0.10‰ (2 s.d., $n = 5$), 0.31 ± 0.12‰ (2 s.d., $n = 5$), 0.25 ± 0.06‰ (2 s.d., $n = 2$), and −0.07 ± 0.03‰ (2 s.d., $n = 5$), respectively, in agreement with previous measured values at the HFUT[50]. Moreover, K from a seawater standard (NASS-5) and two rock reference materials (AGV-2, BHVO-2) was purified following this procedure, giving $δ^{41}K_{NASS-5}$ of +0.13 ± 0.08‰ (2 s.d., $n = 4$), $δ^{41}K_{AGV-2}$ of −0.44 ± 0.11‰ (2 s.d., $n = 7$), and $δ^{41}K_{BHVO-2}$ of −0.52 ± 0.04‰ (2 s.d., $n = 2$), in line with previously published data[50–53]. Overall, the long-term external reproducibility is better than 0.11‰ (2 s.d.) for $δ^{41}K$ measurements[50].

## Data availability
The datasets generated in this study are provided in the Supplementary Information. Source Data is provided with this paper https://doi.org/10.6084/m9.figshare.30665387.

## Code availability
The code used in this manuscript (PHREEQC software) is available for download from the U.S. Geological Survey website: https://www.usgs.gov/software/phreeqc-version-3.

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

## Acknowledgments

This work was financially supported by NSFC (42221003 and 42530512 to Z. Jin; 42373054 to L.-F. Gou; and 42373001 to H. Sun), and ERC Consolidator grant 682760 to PPvS. DJW was supported by a Natural Environment Research Council independent research fellowship (NE/T011440/1), and Key Fund of the State Key Laboratory of Loess and Quaternary Geology (SKLLQGZD2504) for Z. Jin and L.-F. Gou. X.-Y. Zheng, S. Li, J. Wang, F. Zhang, and M. He are thanked for their insightful comments that benefited this manuscript. C. Huang at Hefei University of Technology is thanked for his help during laboratory work.

## Author contributions

Z. Jin and L.-F. Gou conceived and led this project, designed and executed the experiments, and wrote the draft manuscript. P. Pogge von Strandmann, W. Li, D.J. Wilson, J. Xiao, Z.-Q. Zhao, and A. Galy discussed the results and reviewed the manuscript. H. Sun and H. Gu analyzed the samples and discussed the results.

## Competing interests

The authors declare no competing interests.
