## [Transparent Peer Review file · Nature Communications]

K isotopes trace temporal silicate weathering intensity

Corresponding Author: Dr Longfei GOU

Version 0:

Reviewer comments:

Reviewer #1

(Remarks to the Author)

Review of NCOMMS-25-24937

I am pleased to provide a review of this manuscript titled "K isotopes trace spatio-temporal silicate weathering intensity". To my knowledge, this would be the first study where riverine K isotopes are measured in continuously collected samples, strengthening the argument that K isotopes might be effective tracers of silicate weathering/secondary mineral formation in terrestrial environments. The data presented are of sound quality, the trends are overall obvious, and the suggested correlation between $d_{41}K$ and weathering intensity (Fig. 3) is believable and exciting. However, there are several points in the manuscript where the mechanisms controlling K isotope fractionation are incorrectly discussed, and potentially relevant processes controlling the K isotopic composition of these riverine samples are dismissed without proper reasoning. There is also little to no discussion of the "spatio" component of the variability described here, with the paper mainly focusing on temporal trends in riverine K data (this is fine, but the title would need to be revised). Finally, the potential implication of this study to deep-time research on silicate weathering intensity is only quickly glanced over at the end of the paper. While I am in favor of the eventual publication of this manuscript, these shortcomings must be addressed because they significantly weaken the quality of the paper in its present form.

Main comments:

Lines 155-156: This is a very simplistic representation of K isotope effects, or isotope theory in general. I first suggest you change the language. Isotopes don't "prefer" a phase over the other, isotopes are preferentially incorporated into one phase or the other. Second, equilibrium K isotope effects result from changes in coordination number/bond length/bond strength. So, for example, if K is bonded to fewer atoms in a solid than it is in aqueous solution, then the heavy K will be preferentially incorporated into the solid (a helpful reference for K isotopes would be Li et al., 2017, GCA, where they do salt experiments and see varying K isotope effects between mineral and fluid as a function of bond length). Also, the citations provided here are not proper. You should include theoretical work on stable isotope effects, which apply just as much to the K case (e.g., there's a nice summary of factors controlling equilibrium isotope effects in Schauble, 2004 - "Applying stable isotope fractionation theory to new systems") or even the abovementioned Li et al. (2017) GCA paper for a K-specific example.

Lines 219-221: The reader should be given a little more information here on these modelling efforts. For example, highlight that these models only consider thermodynamics and not kinetics, which is known to affect K isotopes. Also, the reader should know why you chose these fractionation factors here. For example, what is the process leading to a K isotope effect of -1.05‰ during Rayleigh incorporation? – I see now that you add this information later, but I suggest it be moved here for clarity and flow. Also, the choice of fractionation factors is questionable based on cited sources.

Lines 230-232: This doesn't make much sense. The ionic radius quoted here refers to K in halides, which does not apply to K adsorbed onto mineral surfaces. Adsorption can occur with hydrated and dehydrated cations and cation affinity is mineral-specific. For example, see this study in which K is seen to be more effectively adsorbed onto montmorillonite than Li (<https://access.onlinelibrary.wiley.com/doi/epdf/10.2136/sssaj1986.03615995005000010008x>)

Lines 237-239: This sentence suggests that K in octahedral sites shows the opposite fractionation to interlayer sites when, in reality, K does not occupy octahedral sites. In fact, K incorporation onto clays goes a bit beyond interlayer vs. adsorption. You might get interlayer K that is fully hydrated and less fractionated relative to the source fluid. Please refer to this study as an example, which is focused on Li but the theory applies to K as well (minus the octahedral sites): Hindshaw et al., 2019, GCA

Lines 239-242: This is not necessarily true. K isotopes could simply be affected by temperature, in addition to other parameters, which would obscure a potential correlation between the two elements based on temperature alone. K also participates in biological uptake, for example (e.g., Higgins et al., 2022, *Frontiers of Physiology*), which doesn't affect Li as much, and it also fractionates quite a lot during kinetic processes like diffusion (e.g., Bourg et al., 2010). Please reconsider this sentence.

Lines 252-253: This range of values is not entirely applicable to this system. Some of these experiments were carried out in ultra-acidic conditions, which are not the case for the Yellow River. Also, the most fractionated K in these experiments happen within minutes (~10 min) of the reaction starting, which I also believe not to be applicable to your system.

Lines 259-262: It's fine to use constraints from your own data but it is not accurate to say that there are no constraints out there for K isotope fractionation during K uptake into secondary silicates. There is no value for lab experiments, but there are constraints from *Ab initio* calculations (e.g., see Zeng et al., 2019, *ACS Earth and Space Chemistry*), measurements of marine sediments and pore-fluids (e.g., Santiago Ramos et al., 2018, *GCA*; your co-author Wenshuai Li's paper from 2022 in *EPSL*), as well as fractionations derived from isotope mass balance calculations (e.g., Li et al., 2019, *PNAS*; Hu et al., 2020, *Science Advances*; Zheng et al., 2022, *EPSL*).

Lines 305-308: This reads as an afterthought and is also inaccurate as it implies the absence of K isotope effects post deposition, when there are now several studies discussing K isotope cycling during marine sediment diagenesis. If you would like to keep a discussion of potential implications for deep time given the results presented here, this thought needs to be revised and properly fleshed out.

Additional comments:

Line 32: Please clarify what your setting is here.

Lines 89-91: There is a significant peak here in end of November/beginning of December. Discharge levels are within Monsoon values, but there doesn't seem to be much of a response in river chemistry. Why is that?

Figure S3 is missing key data sources. There is a large dataset of AOC d41K in Santiago Ramos et al. (2020). Fertilizer and USGS standards were analyzed in Morgan et al. (2018). Please revise your citations.

Lines 149-150: Unclear what the authors mean by this sentence. Please clarify.

Lines 156-158: Unclear what the authors are trying to say here. Are you explaining Rayleigh fractionation? Please clarify.

Lines 171-173: Is there any role for seasonal biogenic K variation in a river system like this one (e.g., Chaudhuri et al., 2007, *Chem Geo*)? I've always been curious about that, given that the d41K of plant material can be quite different from the UCC value.

Lines 200-201: Rather than plant growth, have you considered the role of plant litter to the riverine load?

Lines 223-224: The definition of Na* should be provided here.

Lines 242-244: This does not read like a strong conclusion based on the discussion leading up to it for reasons I highlighted above.

Lines 246-248: Na* should have been explained earlier on when you mention the Rayleigh and batch models in Fig. 2. And the PHREEQC calculations are casually mentioned here, that should not be the case. You need to provide the reader more background than simply model outputs on a supplementary table.

Paragraph starting in line 249: This information should be provided when Fig. 2 is first mentioned. Otherwise, it is hard to assess the choice of fractionation factors, for example, which is crucial to the modelling efforts.

Line 256: Should this be Na*?

Line 298: Please define WI or stay consistent with previous definition of weathering intensity as S/D in lines 280-281.

Lines 350-353: I would suggest adding the caveat that such modeling only considers thermodynamic equilibrium at the exclusion of any kinetic processes likely affecting K partitioning between solid and aqueous phases. Perhaps in the discussion would be more appropriate than here. As is, there is little info in the discussion when you mention these models and that weakens the point being made.

Reviewer #2

(Remarks to the Author)

I am pleased to review the manuscript by Gou et al., "K isotopes trace spatio-temporal silicate weathering intensity". This study explores the application of K isotopes to trace silicate weathering, establishing an empirical relationship between weathering intensity (S/D) and K isotopes using Yellow River (new data) and Yangtze River (literature) datasets. The work offers valuable insights into K isotopes as a weathering proxy. However, several major concerns regarding the interpretation of the K isotope data warrant discussion, detailed below. I hope these comments will be helpful to the authors during revision.

First, on short (seasonal) timescales, silicate weathering might not be the sole control on K isotopes. Other processes, such as contributions from different source rocks (e.g., evaporites, potentially 10-40% of riverine K; Fig. S6) and the mixing of various water sources (rainwater, runoff, shallow/deep groundwater), could also influence seasonal K isotope variability. This mixing, in particular, might generate an "apparent" K isotope-S/D relationship.

Second, the representativeness of the Yellow River system, dominated by loess, for global weathering patterns is questionable. While loess composition may resemble upper continental crust, its erodibility and weatherability likely differ substantially from more globally prevalent rocks like granite and basalt.

Third, the poor fit between the global large river data of Wang et al. (2021) and the study's S/D-K isotope relationship (Fig. 3) is concerning. The authors suggest the global data represent "snapshot sampling," but the same limitation applies to individual samples of this study. Consequently, applying this empirical relationship via marine archives to reconstruct Earth's historical silicate weathering intensity requires further support.

Additional comments:

1. Discrepancy between Fig S6 and Fig S7: Carbonate-sourced K varies 0-40% in S7 but is negligible in S6. Please clarify.
 2. Lines 200-201: Does this imply the S/D-K relationship is invalid in vegetated subtropical/tropical regions?
 3. Lines 242-244: If silicate dissolution and clay precipitation control both Li and K isotopes, why is there no correlation in Fig S10?
- Line 308: Typo: "though" should be "through."

Version 1:

Reviewer comments:

Reviewer #2

(Remarks to the Author)

Gou et al. have presented a revised version of the manuscript "K isotopes trace temporal silicate weathering intensity." In general, the authors have addressed most of the comments I provided in the previous version. Although I am not an expert in K isotopes, the revised version appears to be well-written and the conclusion (the empirical W/D-K isotope relationship) is indeed attractive to a broad community in Earth Science. Prior to its publication in NC, I would encourage the authors to address few more comments below.

Major Comments

1. Writing style. There is too much information in the Supplementary Info (but I agree that these figures/text are indeed necessary), and it may be difficult for readers to switch between the main text and SI. I would suggest that the authors move some of the figures/text to the main text/Methods section. For example, Figures S6 and S7 emphasize the dominance of silicate weathering in the K budget, and one of them should be moved to the main text (there are currently only 3 figures).
2. Saturation index (Lines 296-298 and Table S4). A saturation index of >0 does not necessarily mean the precipitation of a given mineral. In particular, only K-mica shows a SI of >0 , and such a primary mineral should not precipitate during the weathering process – it may simply stop dissolving. Interestingly, why does mica – a very soluble mineral – show a SI of >0 , while K-feldspar – a less soluble mineral – can still dissolve in the river water? Additionally, the clay mineral illite (which is important for the argument) is somehow undersaturated. Although this question does not affect the key conclusion of this manuscript, I would encourage the authors to re-check the PHREEQC calculations.
3. The application of K isotopes in non-biological oceanic authigenic carbonate records. This sounds like a promising future research direction. However, it might be difficult to find a continuous record of non-biogenic authigenic carbonate in the oceans, as most carbonate is biogenic. As such, the authors may need to acknowledge this limitation, or propose a specific geological setting where such a record may exist.

Other Comments

4. In the reply to Question 6 of Reviewer 2, the authors argued that "in low weathering-intensity regimes (W/D close to 0)... whereas the dissolved K isotopes have to be lighter than rock values." However, in the observations (Fig. 3), the K isotope ratio is quite high at very low W/D (Fig. 3). This contrast needs to be reconciled in the revised text.
5. Related to comment 4: Just out of curiosity, why could the release of light K during the initial weathering stage not be observed in the global K isotope dataset?
6. Line 286: "the ab initial calculations³⁶." What does "ab" mean?
7. In the Methods, it is mentioned that 60 river samples were collected. However, there might not be 60 data points in Fig. 3. It may be better to present the exact number of river K isotope data in the figure caption, which may also be a requirement by the journal.
8. For a better understanding, the arrows in Fig. R6 of the rebuttal letter should also be indicated in Fig. S9 of the SI.

Reviewer #3

(Remarks to the Author)

Review of NCOMMS-25-24937A

I am delighted to provide some comments on the manuscript NCOMMS-25-24937A. Gou et al. reported the first seasonal riverine K isotopes, a comprehensive and timely investigation of potassium (K) isotope systematics in the Yellow River, with the aim of exploring how seasonal variations in K isotopes can trace changes in silicate weathering intensity. The study represents an innovative and valuable contribution to the rapidly growing field of nontraditional stable isotope geochemistry and chemical weathering. On the whole, this is a novel and well-written manuscript that stands out for its originality, clarity, and scientific rigor. The authors have successfully integrated field observations, isotopic measurements, and geochemical modeling to solve the two current standing problems that, first, whether there is seasonal K isotope variation since all reported spatial data neglecting seasonal variation; and the second, data from the Yangtze River suggest K isotopes as a tracer of silicate weathering intensity, while global data don't. This research addresses the above two important knowledge gap. They observed clear seasonal K isotopes variation showing that there is huge seasonal riverine K isotopes variation, and they found how properly apply K isotopes as a tracer of weathering intensity, and does so through a carefully designed and well-executed case study on an ideal setting. Though time-financial-consuming via staying in one site for collecting those samples a whole year, such method of decreasing-variable is reasonable and thought-provoking. The discussion effectively connects riverine geochemistry to broader Earth surface processes, making the manuscript not only relevant to isotope geochemists but also to a wide community interested in weathering, erosion, and Earth system feedbacks. In their response to the reviewers' comments, the authors have provided thorough and satisfactory answers to all the questions

raised and have made substantial improvements to the manuscript. I would like to see that such a study represents a significant advancement and is suitable for publication in Nature Communications.

Below are a couple of minor concerns that I encourage the authors to consider revising.

L 48, stabilize, consider revising.

L55-66, the study is among the first to investigate the seasonal variability of K isotopes in a large river system, while K isotopes have recently been explored as tracers of silicate weathering, most prior work has focused on global averages or on spatial catchments. By contrast, this paper presents a temporal dataset covering different seasons, hydrological conditions, and tributaries. This temporal dimension is crucial because silicate weathering intensity fluctuates seasonally with changes in temperature, runoff, and biological activity. However, the authors failed to emphasize the efforts of SEASONAL variation on controlling K isotopes fractionation mechanism. I saw your statement of weak biological control in the following paragraph, it should also be mentioned in the introduction. Further, it is recommended to slightly expand on how K isotope behavior differs from that of other alkali metals such as Li or Na, since comparisons to Li isotopes are frequently made in weathering studies. A brief clarification would further highlight why K isotopes provide unique insights.

L 67-76, The Yellow River sounds an ideal system for this study, but the authors seem failed to present the whole picture of the current state. For example, because of its large size, diverse lithology, strong seasonal hydrological variability, and weak human influence. Insights derived from this system are thus relevant to understanding modern continental weathering processes and their impact on global elemental cycles. I would encourage the authors add more explanation. Such as, by establishing the link between K isotopes and weathering intensity, current work provides a foundation for interpreting ancient K isotope records in marine sediments — potentially offering a new proxy for reconstructing paleo-weathering and paleo-climate conditions.

L 100, characterized, consider revising.

L176, ~95.0% ? attention the significant figures.

L 188-190, the K budget in the atmospheric reservoir appears mostly sourced from the biomass burning and traffic emission (Qu et al., 2024) since K is critical for organisms, I am curiosity if there is also the signal of biomass burning or traffic emission in rivers, I acknowledge that the authors have calculated the atmospheric input to riverine reservoir as less than 1%, which may not be critical to affect the conclusion of the current manuscript, but still interesting to open other door of biological K cycles in hydrosphere.

L 200, fertilizers, consider revising.

L 202-204, the traditional opinion holds that biologically light K is favored, however, more evidences show that there are both light and heavy K isotopes favored by difference vegetation leaves (Qu et al., 2022), please revise. Please explicitly mention the potential role of biological cycling (plant uptake and litter return) in modifying $\delta^{41}\text{K}$, even if it is likely minor in this context.

L239, normalized, consider revising.

L 259-270, the final statement about Li–K decoupling (Fig. S9) is valuable, but there is lacking an explanation, please specify whether Li–K decoupling implies distinct controlling processes or differing sensitivities to diffusion, adsorption, or temperature, because both are proposed as a silicate weathering tracer (Dellinger et al., 2015).

L 276, behavior, consider revising.

L 360-368, The authors end by outlining the future potential of K isotopes as both modern and paleoenvironmental proxies, an exciting and appropriate direction for continued research. Broader implication for the interpretation of K isotopes in geological archives. The results suggest that $\delta^{41}\text{K}$ variations in marine carbonates could record temporal changes in continental weathering intensity, potentially linked to climate variability. The authors appropriately note that this interpretation must account for riverine flux weighting and secondary processes during sediment transport, but the potential of K isotopes as a paleoweathering proxy is compelling. By quantifying the link between hydrology and isotopic composition, this paper offers an important step toward integrating K isotopes into global weathering–climate feedback models. However, the authors should keep in mind that there is rare non-biological carbonates. I would suggest that the authentic clay could be served as the archivers before diagenesis, since the river water (seawater providing that the hydrothermal input keeps table, Zheng et al., 2022).

The final sentence is quite long and dense. Splitting it into two sentences would improve readability, one emphasizing the constancy of the fractionation factor, and another highlighting its application for reconstructing paleo-seawater $\delta^{41}\text{K}$ and weathering feedbacks

L 360, conceptualized, considering revising.

Refs.

Dellinger, M., Gaillardet, J., Bouchez, J., Calmels, D., Louvat, P., Dosseto, A., Gorge, C., Alanoca, L., Maurice, L. (2015). Riverine Li isotope fractionation in the Amazon River basin controlled by the weathering regimes. *Geochimica et Cosmochimica Acta* 164, 71-93.

Qu R. and Han G. (2022). Potassium isotopes in herbaceous plants: A potential new tool for C3 and C4 plant research. *Journal of Geophysical Research: Biogeosciences*, 127, e2021JG006682.

Qu R., Han G. Zeng J. (2024). New tracer for anthropogenic pollution in the atmosphere: Stable potassium isotopes in

rainwater, *Journal of Cleaner Production*, 435, 140574.

Zheng X.-Y., Brian L. B., Mason N., Maria F. F., Julia G. Bryce, C. M. J. (2022). Stable potassium (K) isotope characteristics at mid-ocean ridge hydrothermal vents and its implications for the global K cycle, *Earth and Planetary Science Letters*, 593, 117653.

STATE KEY LABORATORY
OF LOESS SCIENCE

CHANG'AN UNIVERSITY

INSTITUTE OF
EARTH ENVIRONMENT

CHINESE ACADEMY OF
SCIENCES

Detailed responses to reviewer comments on NCOMMS-25-24937

Reviewer comments are in plain text. Our replies are in **blue text**. We have carefully revised the text (marked **in RED** in the revised manuscript) in response to these helpful comments, and we provide the revised manuscript with line numbers (RML) below.

Reviewer #1 (Remarks to the Author):

I am pleased to provide a review of this manuscript titled “K isotopes trace spatio-temporal silicate weathering intensity”. To my knowledge, this would be the first study where riverine K isotopes are measured in continuously collected samples, strengthening the argument that K isotopes might be effective tracers of silicate weathering/secondary mineral formation in terrestrial environments. The data presented are of sound quality, the trends are overall obvious, and the suggested correlation between $d_{41}K$ and weathering intensity (Fig. 3) is believable and exciting. However, there are several points in the manuscript where the mechanisms controlling K isotope fractionation are incorrectly discussed, and potentially relevant processes controlling the K isotopic composition of these riverine samples are dismissed without proper reasoning. There is also little to no discussion of the “spatio” component of the variability described here, with the paper mainly focusing on temporal trends in riverine K data (this is fine, but the title would need to be revised). Finally, the potential implication of this study to deep-time research on silicate weathering intensity is only quickly glanced over at the end of the paper. While I am in favor of the eventual publication of this manuscript, these shortcomings must be addressed because they significantly weaken the quality of the paper in its present form.

Re: Accepted. We revised our discussion of the mechanisms controlling K isotope fractionation and potential processes controlling the K isotopic composition of these samples. As rightly pointed out, this work focuses on the seasonal variations in riverine K isotopes and then extends to global “spatio” component since we included the published “spatio” data, so we revised the title according to your suggestion to “K isotopes trace temporal silicate weathering intensity”. We added the potential implications of this study to deep-time research on silicate weathering intensity to strengthen the impact of the paper.

Main comments:

1. Lines 155-156: This is a very simplistic representation of K isotope effects,

or isotope theory in general. I first suggest you change the language. Isotopes don't "prefer" a phase over the other, isotopes are preferentially incorporated into one phase or the other. Second, equilibrium K isotope effects result from changes in coordination number/bond length/bond strength. So, for example, if K is bonded to fewer atoms in a solid than it is in aqueous solution, then the heavy K will be preferentially incorporated into the solid (a helpful reference for K isotopes would be Li et al., 2017, GCA, where they do salt experiments and see varying K isotope effects between mineral and fluid as a function of bond length). Also, the citations provided here are not proper. You should include theoretical work on stable isotope effects, which apply just as much to the K case (e.g., there's a nice summary of factors controlling equilibrium isotope effects in Schauble, 2004 - "Applying stable isotope fractionation theory to new systems") or even the abovementioned Li et al. (2017) GCA paper for a K-specific example.

Re: Accepted. We revised the manuscript following your valuable comment. Specifically, we changed the language to "heavy K isotopes are preferentially incorporated into the evaporite phase due to the equilibrium K isotope effects that result from changes in coordination number/bond length/bond strength of K-bearing evaporites". We also properly cited the references of Li et al. (2017) and Schauble (2004). RML155-161.

2. Lines 219-221: The reader should be given a little more information here on these modelling efforts. For example, highlight that these models only consider thermodynamics and not kinetics, which is known to affect K isotopes. Also, the reader should know why you chose these fractionation factors here. For example, what is the process leading to a K isotope effect of -1.05‰ during Rayleigh incorporation? – I see now that you add this information later, but I suggest it be moved here for clarity and flow. Also, the choice of fractionation factors is questionable based on cited sources.

Re: Accepted. Thanks for your helpful feedback. We clarified that these models are thermodynamic models (RML 225-228). We moved the key information about the fractionation factors to here for clarity and flow (RML 229-249).

We also checked the fractionation factors and the references, as further discussed in your comment #6 (below). We completely agree with you that there may be some uncertainties associated with the difference between laboratory experiments and field studies, but we felt they are the most relevant parameters that we could obtain so far.

3. Lines 230-232: This doesn't make much sense. The ionic radius quoted here refers to K in halides, which does not apply to K adsorbed onto mineral surfaces. Adsorption can occur with hydrated and dehydrated cations and cation affinity is mineral-specific. For example, see this study in which K is seen to be more effectively adsorbed onto montmorillonite than

Li

(<https://access.onlinelibrary.wiley.com/doi/epdf/10.2136/sssaj1986.03615995005000010008x>)

Re: Accepted. We removed these statement and references from the manuscript. Then we revised the manuscript according to your helpful suggestions. RML276-287.

4. Lines 237-239: This sentence suggests that K in octahedral sites shows the opposite fractionation to interlayer sites when, in reality, K does not occupy octahedral sites. In fact, K incorporation onto clays goes a bit beyond interlayer vs. adsorption. You might get interlayer K that is fully hydrated and less fractionated relative to the source fluid. Please refer to this study as an example, which is focused on Li but the theory applies to K as well (minus the octahedral sites): Hindshaw et al., 2019, GCA

Re: Accepted. We revised this sentence and cited the relevant literature. "Nuclear magnetic resonance (NMR) spectroscopy shows the evidence for Li but applicable to K that outer-sphere K appears fully hydrated and less fractionated relative to the source fluid (Hindshaw et al., 2019), together with experiments show..." RML276-280.

5. Lines 239-242: This is not necessarily true. K isotopes could simply be affected by temperature, in addition to other parameters, which would obscure a potential correlation between the two elements based on temperature alone. K also participates in biological uptake, for example (e.g., Higgins et al., 2022, Frontiers of Physiology), which doesn't affect Li as much, and it also fractionates quite a lot during kinetic processes like diffusion (e.g., Bourg et al., 2010). Please reconsider this sentence.

Re: Accepted and revised. 1) We removed the explanation about the correlation between Li and K and the associated description. 2) We reinforced the evidence for limited biological uptake and release of K in the Yellow River. 3) The diffusion of K in liquid water has $D \propto m^{-\beta}$ with $0 \leq \beta < 0.20$, where D , m , and β refer to the diffusion coefficient, the mass of the diffusing particle, and the mass-scaling exponent, respectively (Bourg et al., 2010). Therefore, at the molecular dynamics scale, to the residence time of water molecules in the Yellow River solute's first solvation shell, we could predict that in hydrodynamic modes is weakened in the dry season, the fractionation of stable isotopes should be enhanced, given the inferred mass dependent diffusion control on the Li isotopes variations sensitive to temperature (Gou et al., 2019), (i.e. heavier K isotopes in water), which is contrary to our observations. RML259-270.

6. Lines 252-253: This range of values is not entirely applicable to this system. Some of these experiments were carried out in ultra-acidic conditions, which are not the case for the Yellow River. Also, the most

fractionated K in these experiments happen within minutes (~10 min) of the reaction starting, which I also believe not to be applicable to your system.

Re: Accepted. We checked the literature, though this experiment conducted in acidic conditions, which indeed provided us the caveat. In the revision, we expanded the fractionation factor to 1.00000, allowing a wider range of possibilities to be covered. Further, the selection of specific fractionation factors appears not to affect our overall conclusion of clay formation. RML 229-241.

7. Lines 259-262: It's fine to use constraints from your own data but it is not accurate to say that there are no constraints out there for K isotope fractionation during K uptake into secondary silicates. There is no value for lab experiments, but there are constraints from Ab initio calculations (e.g., see Zeng et al., 2019, ACS Earth and Space Chemistry), measurements of marine sediments and pore-fluids (e.g., Santiago Ramos et al., 2018, GCA; your co-author Wenshuai Li's paper from 2022 in EPSL), as well as fractionations derived from isotope mass balance calculations (e.g., Li et al., 2019, PNAS; Hu et al., 2020, Science Advances; Zheng et al., 2022, EPSL).

Re: Accepted, we revised the manuscript following your suggestion. We also carefully checked the constraints from Ab initio calculations, and natural observations. RML241-249.

8. Lines 305-308: This reads as an afterthought and is also inaccurate as it implies the absence of K isotope effects post deposition, when there are now several studies discussing K isotope cycling during marine sediment diagenesis. If you would like to keep a discussion of potential implications for deep time given the results presented here, this thought needs to be revised and properly fleshed out.

Re: Accepted. We tried our best to provide the potential implications for deep time. We revised this part and added several sentences to describe the K isotope effects post deposition according to marine sediment diagenesis. Although global seawater $\delta^{41}\text{K}$ (~+0.12 ‰) is higher than UCC (~-0.44 ‰), which is mainly proposed to be attributable to sediment sinks and early diagenesis (Wang et al., 2020; Li et al., 2022b), rather than terrestrial weathering input (-0.38±0.04‰ (Wang et al., 2021)) and mid-ocean ridge vent system release (-0.46‰ and -0.15‰ (Zheng et al., 2022)), existing data for terrestrial weathering input of K isotope data lack constraints on seasonal variability (Li et al., 2019; Wang et al., 2021). Here we show that the riverine K isotopes could be as heavy as 0.27‰ during the storm event under incongruent weathering conditions with a large amount of fresh mineral supply and aluminosilicate neoformation. If the global rivers all show K isotope seasonality, terrestrial weathering could fully explain the seawater K isotope evolution without any additional endmember changes. In deep time, since biogenic carbonates appear to be affected by biological fractionation

of K isotopes (Li et al., 2022a), the non-biological oceanic authigenic carbonates with a constant (likely temperature-dependent) fractionation factor between seawater and carbonates could serve as an archive of paleo-seawater K isotopes. RML 340-368.

Additional comments:

9. Line 32: Please clarify what your setting is here.

Re: Accepted. We clarified the setting to make the manuscript clear.

We therefore investigate the seasonal dissolved K isotopes in the middle Yellow River, which is covered by the homogeneous loess that can represent the average geochemical composition of the Upper Continental Crust (UCC), and is influenced by significant climatic seasonality driven by the East Asian monsoon. Our setting in the middle Yellow River is ideal to assess the K isotope response to the climatic parameters forcing silicate weathering. RML 30-38.

10. Lines 89-91: There is a significant peak here in end of November/beginning of December. Discharge levels are within Monsoon values, but there doesn't seem to be much of a response in river chemistry. Why is that?

Re: Insightful comments. We noticed that the concentration of major ions is negatively correlated with discharge, probably related to precipitation and groundwater circulation (Fig. 1; Gou et al., 2023).

11. Figure S3 is missing key data sources. There is a large dataset of AOC d41K in Santiago Ramos et al. (2020). Fertilizer and USGS standards were analyzed in Morgan et al. (2018). Please revise your citations.

Re: Accepted. Thanks for pointing out our omissions. We have now revised the figure and cited the relevant references. See revised Figure S3.

12. Lines 149-150: Unclear what the authors mean by this sentence. Please clarify.

Re: Nice feedback. Originally, we meant to point out that Li et al. (2019) and Wang et al. (2022) both argued that there is no seasonal variation of K isotopes observed in the Yangtze and global rivers. We have now revised this sentence to better clarify our meaning. RML 149-151.

14. Lines 156-158: Unclear what the authors are trying to say here. Are you explaining Rayleigh fractionation? Please clarify.

Re: Helpful feedback. We revised this sentence. The evaporites and secondary carbonates in loess have been formed after the primary eolian loess deposition. Since heavy K isotopes are preferentially incorporated into the evaporite phase due to the equilibrium K isotope effects resulting from changes in coordination number/bond length/bond strength of K-bearing

evaporites, higher $\delta^{41}\text{K}$ values of evaporites and secondary carbonates are expected than that of the silicates in loess, as observed. RML155-161.

15. Lines 171-173: Is there any role for seasonal biogenic K variation in a river system like this one (e.g., Chaudhuri et al., 2007, Chem Geo)? I've always been curious about that, given that the $\delta^{41}\text{K}$ of plant material can be quite different from the UCC value.

Re: Insightful concerns. Plants are enriched in light K isotopes (Christensen et al., 2018). Here we excluded the role of seasonal biogenic K variation in the middle Yellow River based on the following evidence. First, vegetation is sparse in the (semi-)arid middle Yellow River. Second, after the ice-melting period, plant growth is enhanced, while the most negative riverine K isotopes occurred at this time, and there are similar K isotope values during both the ice-melting interval and the monsoon season (Fig.1). The K isotopes are positively correlated with the dissolved K flux and the chemical weathering rate (Fig. S8), suggesting silicate dissolution control on the dissolved K budget, because evaporite-sourced K should only be sensitive to discharge rather than chemical weathering rate. Together with the absence of source mixing relationships (Fig. S7) and negligible carbonate-sourced K (Fig. S6), the riverine K isotopes should predominantly reflect natural weathering processes related to K release from silicates and K uptake by SPM. RML 200-217.

16. Lines 200-201: Rather than plant growth, have you considered the role of plant litter to the riverine load?

Re: Thanks for the suggestion, and yes. We considered plant litter supply alongside plant growth. We ruled out this possibility for three reasons. 1) The plants are sparse in such a (semi-)arid region, so their defoliation should not be impactful; 2) After the monsoon season, the K/Sr ratios remain stable (even slightly decrease), whereas if the plant litter contribution was significant, we should see the rise of K/Sr ratio because K is input by plant defoliation (Fig. R1). 3) Plant defoliation occurs in October, while the K isotopes in this period are even higher than that in the spring (Fig. R1). In contrast, we see the lowest K/Sr during the storm event, corresponding to the heaviest K isotopes coincidentally, implying the weathering processes control on K isotopes (Fig. R1).

Fig R1 The K/Sr ratio from January to December in 2013.

17. Lines 223-224: The definition of Na* should be provided here.

Re: Accepted and defined. RML 240-241.

18. Lines 242-244: This does not read like a strong conclusion based on the discussion leading up to it for reasons I highlighted above.

Re: Accepted, removed and revised. RML 259-270.

19. Lines 246-248: Na* should have been explained earlier on when you mention the Rayleigh and batch models in Fig. 2. And the PHREEQC calculations are casually mentioned here, that should not be the case. You need to provide the reader more background than simply model outputs on a supplementary table.

Re: Accepted. Na* has been defined above, RML 223-224. The PHREEQC calculations are revised in the supplementary materials, with further background provided in the main text. RML 289-298.

20. Paragraph starting in line 249: This information should be provided when Fig. 2 is first mentioned. Otherwise, it is hard to assess the choice of fractionation factors, for example, which is crucial to the modelling efforts.

Re: Accepted and revised. RML 229-249.

21. Line 256: Should this be Na*?

Re: Yes, this is a typo, revised as Na*. RML239.

22. Line 298: Please define WI or stay consistent with previous definition of weathering intensity as S/D in lines 280-281.

Re: Accepted. We decided to stay consistent with previous definition of weathering intensity as W/D for the convenience of discussion and comparison purpose. See revised Fig. 3.

23. Lines 350-353: I would suggest adding the caveat that such modeling only considers thermodynamic equilibrium at the exclusion of any kinetic processes likely affecting K partitioning between solid and aqueous phases. Perhaps in the discussion would be more appropriate than here. As is, there is little info in the discussion when you mention these models and that weakens the point being made.

Re: Accepted. We added the caveat that such modeling only considers thermodynamic equilibrium at the exclusion of any kinetic processes likely affecting K partitioning between solid and aqueous phase in the discussion. It indeed strengthened our point. RML225-228 & RML 289-298.

Reviewer #2 (Remarks to the Author):

I am pleased to review the manuscript by Gou et al., “K isotopes trace spatio-temporal silicate weathering intensity”. This study explores the application of K isotopes to trace silicate weathering, establishing an empirical relationship between weathering intensity (S/D) and K isotopes using Yellow River (new data) and Yangtze River (literature) datasets. The work offers valuable insights into K isotopes as a weathering proxy. However, several major concerns regarding the interpretation of the K isotope data warrant discussion, detailed below. I hope these comments will be helpful to the authors during revision.

Re: We are grateful for your very constructive comments that helped to significantly strengthen our manuscript and hopefully create a more impactful paper.

1. First, on short (seasonal) timescales, silicate weathering might not be the sole control on K isotopes. Other processes, such as contributions from different source rocks (e.g., evaporites, potentially 10-40% of riverine K; Fig. S6) and the mixing of various water sources (rainwater, runoff, shallow/deep groundwater), could also influence seasonal K isotope variability. This mixing, in particular, might generate an “apparent” K isotope-S/D relationship.

Re: We acknowledge your concern about the mixing but do not consider it to be the main driver of the observed changes for the following reasons.

1) We quantitatively calculated the contribution from each endmember. Compared to the 75% average contribution from silicates, the average contribution from evaporites is only ~25% and hence not the dominant control. In order to remove the effect of evaporite contributions, we only focus on the silicate weathering (see supplementary materials).

2) We assessed the relationship between K isotopes and the

contributions of evaporite, carbonate, and silicate, and all show no correlation (Fig. S7). Further, the average K isotope composition of evaporites is 0.03 (± 0.30), while for silicates it is -0.36 (± 0.12) (Table S3). The evaporite contribution in the dry season is generally higher (Fig. S6), while the riverine K isotopes are lower (Fig. 1), and vice versa, which is opposite to what would be expected if evaporite contributions were driving the signal.

3) The rainwater, runoff, and shallow/deep groundwater are ultimately sourced from the modern and paleo rainwater, but there is very low K in rainwater compared to river water (Table S2), so the K contribution is derived from chemical weathering. We use the dissolved and suspended particulate matter from the river water to calculate the rate and intensity of chemical weathering for each point in time, regardless of the origin of the water. The clear relationship between S/D and K isotopes indicates a control through weathering. However, we recognise that part of these changes could represent contributions from runoff versus groundwater, with different water-rock interaction times. Furthermore, the K isotopes of the groundwater is $-0.05 \pm 0.00\text{‰}$, which is comparable to the annual-average riverine $\delta^{41}\text{K}$ values in the middle Yellow River, and thus unlikely the result of the seasonal K isotope variation.

In summary, 1) K is predominantly sourced from silicates; 2) K is fractionated during the silicate weathering processes of dissolution and clay formation; 3) K isotopes are observed to be controlled by the ratio between soluble and solid phases; and 4) K isotopes should be theoretically correlated to S/D. RML 299-321.

2. Second, the representativeness of the Yellow River system, dominated by loess, for global weathering patterns is questionable. While loess composition may resemble upper continental crust, its erodibility and weatherability likely differ substantially from more globally prevalent rocks like granite and basalt.

Re: We acknowledge these concerns about the representativeness of loess, but we feel our study is representative at a global scale for several reasons.

(1) The erodibility and weatherability of loess show no systematic deviation from other non-loess dominated settings on the surficial Earth (Table R1), though erodibility has long been a hot debated topic in geosciences. Meanwhile, different rock types with different mineralogies (granite, andesite, basalt) all show distinct weatherability (Teng et al., 2020). In addition, the high erodibility of loess enables seasonal changes in erosion rate and weathering fluxes of the Yellow River to span a large range, while the source rock remains consistent and represents an integrated average geochemical and mineralogical composition of the upper continental crust, which largely removes lithological controls. It is therefore a conducive setting for observing the impact of large seasonal changes in

W/D on K isotopes, and thereby testing the reliability of K isotopes for tracing W/D.

Table R1 Erosion rates summarized from the literature.

Sites	Short-term erosion rate (t/km ² /yr)	Long-term erosion rate (t/km ² /yr)	References ^a
Himalaya (upper Ganges River)	2120-7155 ^b	2120-7950	Vance et al., 2003
Sierra Nevada, USA	39.75-159 ^b	66.25-159	Granger et al., 2001; Riebe et al., 2001
Idaho Batholith, USA	53-265 ^b	265	Kirchner et al., 2001
Sri Lanka	13.25-29.15 ^b	10.6-39.8	von Blanckenburg et al., 2004
Smoky Mtns. USA	37.1-98.05 ^b	26.5-106	Matmon et al., 2003
Blue Mtns. Australia	26.5-53 ^b	<37.1	Wilkinson et al., 2005
Bolivian Andes	107-3577.5 ^b	265-1590	Safran et al., 2005
Loess/Poland	120-1090 ^b	44-85	Loba et al., 2021
Loess/China	510-4070 ^c	/	Liu et al., 2022

- a. The average density used for the surface materials is 2.65 t/m³ to calculate the m/Myr data in literatures to t/km²/yr for comparison purpose.
- b. Measured by ¹⁰Be isotopes.
- c. Measured by plutonium isotopes.

(2) There is no data showing that the loess weatherability is differ substantially from the other global rivers. Our evidence show that Li, Sr, Ba, and Mg isotopes in the Yellow River are all within the range of the reported global rivers (Figs. R2 and R3) (Gou et al., 2019; Gou et al., 2020; Gou et al., 2023), while K isotopes show the same relationship with W/D as seen in the Yangtze River and other global large rivers (Fig. 3). These observations support the choice of loess and the Yellow River a representative setting for investigating globally relevant seasonal riverine K isotopes variations.

Fig. R2 Li isotopic composition of the Yellow River (far right) is in the range of global rivers, modified from Gou et al. (2017).

Fig. R3 Ba isotopes from the Yellow River show the same slope when plotted with the silicate weathering rate (SWR) to the Mackenzie (unpublished data).

(3) In addition, loess covers 10% of the Earth's surface (Fenn and Prud'Homme, 2022; Fig. R4), so the chemical weathering of loess is by itself an important carbon sequestration process. Furthermore, loess has been proposed as an archive in which to explore weathering, low temperatures weakening chemical weathering and high temperatures strengthening chemical weathering and high temperatures strengthening

chemical weathering (Fenn and Prud'Homme, 2022), making insights into loess weathering valuable.

Fig. R4 Loess distribution worldwide (Fenn and Prud'Homme, 2022)

(4) Our bulk silicate earth (BSE) is composed of thousands of minerals (one may very different from other), actually, the granite, andesite, and basalt all show distinct weatherability (Teng et al., 2020), plus the long-term timescale of chemical weathering, it is nearly impossible to investigate each mineral one by one, and not necessarily to do that because the whole processes of chemical weathering is not the simple sum of each mineral. Alternatively, we put them together and try to select large river which provide us the averaged clues to understanding silicate weathering. Therefore, the loess shows indispensable role for providing the averaged weathering information due to the fact that loess is the homogeneous aeolian physical-weathered deposit (loess contains of all minerals known from the upper continental crust) that could represent the average geochemical composition of the upper continental crust. So, the choice of loess is apparently better than any other single mineral in such settings.

In summary, the representativeness of the Yellow River system, dominated by loess, for global weathering patterns appears ideal. Loess not only geochemically represents upper continental crust, but also physically (i.e. the erodibility and weatherability) represents globally prevalent rocks and minerals.

3. Third, the poor fit between the global large river data of Wang et al. (2021) and the study's S/D-K isotope relationship (Fig. 3) is concerning. The authors suggest the global data represent "snapshot sampling," but the same limitation applies to individual samples of this study. Consequently, applying this empirical relationship via marine archives to reconstruct Earth's historical silicate weathering intensity requires further support.

Re: We acknowledge the concerns about the empirical relationship between S/D and K isotopes, and have adjusted our paper accordingly, but we should clarify that the snapshot limitation does not apply equally to our study as to previous global studies. Our data compares seasonally varying K isotopes with weathering rates and weathering intensities also measured on the same timescales. In contrast, previous limited sampling of global rivers showed snapshot-sampled K isotopes compared to more long-term averaged silicate weathering intensities. The global spatial data represent “snapshot sampling” due to the fact that there is indeed seasonal K isotope variation (Fig. R5). The initial effort of this study is that Wang et al. (2022) and Li et al. (2018) argued that there is no seasonal K isotope variation (Fig. R5). Even in their own data, there is apparent K isotopic variation (at least to the level of 0.2-0.3‰, Fig. R5). Our seasonal sampling strategy efforts to eliminate the limitation of snapshot sampling by decreasing the uncertainty of the seasonal K isotope variation (Fig. R5) and thus establish a vivid panorama of S/D-K relationship. We admit that there must be more work on a range of modern river systems before this empirical relationship is confidently applied via marine archives to reconstruct Earth’s historical silicate weathering intensity, and have added a note to this effect in the paper. RML 341-344.

Figure 2
 Fig. R5 Seasonal K isotope variation in (A) global large rivers; (B) Yangtze River.

Additional comments:

4. Discrepancy between Fig S6 and Fig S7: Carbonate-sourced K varies 0-40% in S7 but is negligible in S6. Please clarify.

Re: Thanks for this very valuable feedback. We checked the excel and found that due to the delete of a column during the calculation that made the data wrongly used in Fig S7. We revised the Fig. S7 to clarify the consistence between Figs. S6 and S7. See the revised Fig. S7.

5. Lines 200-201: Does this imply the S/D-K relationship is invalid in vegetated subtropical/tropical regions?

Re: We appreciate your critical comments. If we take a look on Wang et al. (2022) and Li et al. (2018), those samples from subtropical/tropical in the broad line between S/D-K. The only problem is that both might have ignored the seasonal K isotope variation. This issue is very important because chemical weathering is suggested controlled by lithology, tectonic, climate, and hydrology. The spatial sampling strategy inevitably includes all above factors, so the relationship between S/D-K may have been obscured. Further, seasonal sampling strategy includes climate and hydrology. If we see spatial K isotope variation, there must be seasonal K isotope variation, which could be useful for quantifying the role of climate and hydrology by setting the lithology and tectonic stable. Therefore, we propose that there might be lithology and tectonic parameters obscured the relationship between S/D-K. That is also the reason why we investigate the first seasonal K isotope variation. Back to the vegetated subtropical/tropical regions, with the data from the Amazon and the Yangtze Rivers, they are all in line to S/D-K relationship. We thus are confident that the S/D-K relationship is also valid in vegetated subtropical/tropical regions. See the revised Fig. 3.

6. Lines 242-244: If silicate dissolution and clay precipitation control both Li and K isotopes, why is there no correlation in Fig S10?

Re: Thanks for this insightful comment.

Decoupling of the two systems is due to 3 reasons during dissolution, transportation, and incorporation of silicate chemical weathering. (1) the difference between Li isotopes and K isotopes during mineral dissolution processes. Li isotopes do not fractionate during dissolution, whereas light K isotopes preferentially enter the fluid phase during dissolution. Therefore, in low weathering-intensity regimes (W/D close to 0), where abundant fresh minerals incongruently dissolve into waters, the dissolved Li isotopes are similar to rock values (because there is no fractionation of Li during dissolution), whereas the dissolved K isotopes have to be lighter than rock values (because K isotopes fractionate in the dissolution stage). In high weathering-intensity regimes (W/D close to 1), where there is low clay formation (because clays re-dissolve), then there is limited K and Li isotopic fractionation, leading to dissolved K and Li isotopes that are similar to rock/loess. (2) Li isotopes theoretically are sensitive to diffusion (Bourg et al., 2010) because of its large mass difference (16.7%; the largest among all metal stable isotopes). In ambient condition, liquid water shows that $D \propto m^{-\beta}$ with $0 \leq \beta < 0.20$ (Bourg et al., 2010), where D , m , and β refer to the diffusion coefficient, the mass of the diffusing particle, and the mass-scaling exponent, respectively. Therefore, at the molecular dynamics scale, to the residence time of water molecules in the Yellow River solute's first solvation shell, we could predict that in the dry seasons (when hydrodynamic mode is weakened), the fractionation of Li isotopes should be enhanced (Bourg et al., 2010), leading to heavier Li isotopes in the water, as observed (Gou et al.,

2019). (3) Since Li is a trace element but K is a major element on the UCC, Li rarely forms independent mineral but K usually forms several minerals (e.g. mica, one of the rock-forming minerals; Table S4). Therefore, the incorporation rates, fractionation for Li and K isotopes must be different. As a result, the two systems are decoupled (Fig. 5B).

Fig. R6 Cross plot between Li isotopes and K isotopes.

Line 308: Typo: “though” should be “through.”

Re: Revised. RML 363.

Acknowledgment: We are grateful to two anonymous reviewers and editor Rebecca Neely, and helps from Dr. Xin-Yuan Zheng, and Shilei Li during the revision, for their valuable constructive suggestions that substantially strengthened our manuscript.

References mentioned in the replies above:

Bourg, I.C., Richter, F.M., Christensen, J.N. and Sposito, G. (2010) Isotopic mass dependence of metal cation diffusion coefficients in liquid water. *Geochimica et Cosmochim Acta* 74, 2249-2256.

Christensen, J.N., Qin, L., Brown, S.T. and DePaolo, D.J. (2018) Potassium and Calcium Isotopic Fractionation by Plants (Soybean [*Glycine max*], Rice [*Oryza sativa*], and Wheat [*Triticum aestivum*]). *ACS Earth and Space Chemistry* 2, 745-752.

Fenn, K. and Prud’Homme, C. (2022) Dust deposits: Loess, *Treatise on Geomorphology*, 320-365.

Gou, L.-F., Jin, Z., Galy, A., Gong, Y.-Z., Nan, X.-Y., Jin, C., Wang, X.-D., Bouchez, J., Cai, H.-M., Chen, J.-B., Yu, H.-M. and Huang, F. (2020) Seasonal riverine barium isotopic variation in the middle Yellow River: Sources and fractionation. *Earth and Planetary Science Letters* 531, 115990.

Gou, L.-F., Jin, Z., Galy, A., Xu, Y., Xiao, J., Yang, Y., Bouchez, J., Pogge von Strandmann, P.A.E., Jin, C., Yang, S. and Zhao, Z.-Q. (2023) Seasonal Mg isotopic variation in the middle Yellow River: Sources and fractionation. *Chemical Geology* 619. 121314.

Gou, L.-F., Jin, Z., Pogge von Strandmann, P.A.E., Li, G., Qu, Y., West, A.J., Xiao, J., Deng, L. and Galy, A. (2019) Lithium isotopes in the middle Yellow River: Seasonal variability, sources and fractionation. *Geochimica et Cosmochimica Acta*. 248, 88-108.

Gou, L.-F., Jin, Z. and He, M.Y. (2017) Using lithium isotopes traces continental weathering: Progresses and challenges. *Journal of Earth Environment* 8, 89-102. (In Chinese with English Abstract)

Granger, D.E. and Muzikar, P.F. (2001) Dating sediment burial with cosmogenic nuclides: Theory, techniques, and limitations. *Earth and Planetary Science Letters* 188, 269–281.

Hindshaw, R.S., Tosca, R., Gout, T.L., Farnan, I., Tosca, N.J. and Tipper, E.T. (2019) Experimental constraints on Li isotope fractionation during clay formation. *Geochimica et Cosmochim Acta* 250, 219-237.

Kirchner, J.W., Finkel, R.C., Riebe, C.S. (2001) Mountain erosion over 10 yr, 10 k.y., and 10 m.y. time scales. *Geology* 29, 591–594.

Li, S., Li, W., Beard, B.L., Raymo, M.E., Wang, X., Chen, Y. and Chen, J. (2019) K isotopes as a tracer for continental weathering and geological K cycling. *Proceedings of the National Academy of Sciences of the United States of America* 116, 8740-8745.

Li, W., Liu, X.-M., Wang, K., Hu, Y., Suzuki, A. and Yoshimura, T. (2022a) Potassium incorporation and isotope fractionation in cultured scleractinian corals. *Earth and Planetary Science Letters* 581, 117393.

Li, W., Liu, X.-M., Wang, K., McManus, J., Haley, B.A., Takahashi, Y., Shakouri, M. and Hu, Y. (2022b) Potassium isotope signatures in modern marine sediments: Insights into early diagenesis. *Earth and Planetary Science Letters* 599. 117849.

Liu, Y., Hou, X., Qiao, J., Zhang, W., Fang, M. and Lin, M. (2023) Evaluation of soil erosion rates in the hilly-gully region of the Loess Plateau in China in the past 60 years using global fallout plutonium, *Catena*, 220, 106666. <https://doi.org/10.1016/j.catena.2022.106666>.

Loba, A., Waroszewski, J., Tikhomirov, D., Calitri, F., Christl, M., Sykuła, M. and Egli, M. (2021) Tracing erosion rates in loess landscape of the Trzebnica Hills (Poland) over time using fallout and cosmogenic nuclides. *Journal of Soils and Sediments*, 21, 2952–2968. <https://doi.org/10.1007/s11368-021-02996-x>.

Matmon, A., Bierman, P. R., Larsen, J., Southworth, S., Pavich M., and Caffee M. (2003) Temporally and spatially uniform rates of erosion in the southern Appalachian Great Smoky Mountains. *Geology* 31, 155–158.

Safran, E. B., Bierman, P. R., Aalto, R., Dunne, T., Whipple, K. X., and Caffee, M. W. (2005) Erosion rates driven by channel network incision in the Bolivian Andes. *Earth Surface Processes and Landforms* 30, 1007–1024.

Teng, F.-Z., Hu, Y., Ma, J.-L., Wei, G.-J. and Rudnick, R.L. (2020) Potassium isotope fractionation during continental weathering and implications for global K isotopic balance. *Geochimica et Cosmochimica Acta* 278, 261-271.

Vance, D., Bickle, M., Ivy-Ochs, S., and Kubik, P.W. (2003) Erosion and exhumation in the Himalaya from cosmogenic isotope inventories of river sediments. *Earth and Planetary Science Letters* 206, 273–288.

von Blanckenburg, F., Hewawasam, T., and Kubik, P.W. (2004) Cosmogenic nuclide evidence for low weathering and denudation in the wet tropical highlands of Sri Lanka. *Journal of Geophysical Research* 109, F03008.

Wang, K., Close, H.G., Tuller-Ross, B. and Chen, H. (2020) Global average potassium isotope composition of modern seawater. *ACS Earth and Space Chemistry* 4, 1010-1017.

Wang, K., Peucker-Ehrenbrink, B., Chen, H., Lee, H. and Hasenmueller, E.A. (2021) Dissolved potassium isotopic composition of major world rivers. *Geochimica et Cosmochimica Acta* 294, 145-159.

Wilkinson, M. T., Chappell, J., Humphreys, G.S., Fifield, L.K., Smith, B., and Hesse, P. (2005) Soil production in heath and forest, Blue Mountains, Australia: Influence of lithology and paleoclimate. *Earth Surface Processes and Landforms* 30, 923–934.

Zheng, X.-Y., Beard, B.L., Neuman, M., Fahnestock, M.F., Bryce, J.G. and Johnson, C.M. (2022) Stable potassium (K) isotope characteristics at mid-ocean ridge hydrothermal vents and its implications for the global K cycle. *Earth and Planetary Science Letters* 593. 117653.

STATE KEY LABORATORY
OF LOESS SCIENCE
CHANG'AN UNIVERSITY

INSTITUTE OF
EARTH ENVIRONMENT
CHINESE ACADEMY OF
SCIENCES

Detailed responses to reviewer comments on NCOMMS-25-24937A

Reviewer comments are in plain text. Our replies are in **blue text**. We have carefully revised the text (marked **in RED** in the revised manuscript) in response to these helpful comments, and we provide the revised manuscript line numbers (RML) below.

Reviewer #2 (Remarks to the Author)

Gou et al. have presented a revised version of the manuscript “K isotopes trace temporal silicate weathering intensity.” In general, the authors have addressed most of the comments I provided in the previous version. Although I am not an expert in K isotopes, the revised version appears to be well-written and the conclusion (the empirical W/D-K isotope relationship) is indeed attractive to a broad community in Earth Science. Prior to its publication in NC, I would encourage the authors to address few more comments below.

Major Comments

1. Writing style. There is too much information in the Supplementary Info (but I agree that these figures/text are indeed necessary), and it may be difficult for readers to switch between the main text and SI. I would suggest that the authors move some of the figures/text to the main text/Methods section. For example, Figures S6 and S7 emphasize the dominance of silicate weathering in the K budget, and one of them should be moved to the main text (there are currently only 3 figures).

Re: Accepted. We moved Figures S6 and S12 into the main text; as new

Figures 2 and Figure 4. Now, there are 5 figures in the revised version, aiming to enhance the readability and coherence as the reviewer suggested.

2. Saturation index (Lines 296-298 and Table S4). A saturation index of >0 does not necessarily mean the precipitation of a given mineral. In particular, only K-mica shows a SI of >0 , and such a primary mineral should not precipitate during the weathering process – it may simply stop dissolving. Interestingly, why does mica – a very soluble mineral – show a SI of >0 , while K-feldspar – a less soluble mineral – can still dissolve in the river water? Additionally, the clay mineral illite (which is important for the argument) is somehow undersaturated. Although this question does not affect the key conclusion of this manuscript, I would encourage the authors to re-check the PHREEQC calculations.

Re: We have double checked all saturation index (SI) calculations in PHREEQC (version 3), with measured pH, temperature, and ionic concentrations as inputs. We agree that $SI > 0$ does not guarantee precipitation, but only thermodynamic saturation. Our updated analyses confirm the followings:

1. We recognize that K-mica shows $SI > 0$, reflecting that river water is close to equilibrium with respect to K-mica. Because K-mica is a primary mineral and relatively easy to be dissolved, so that it could have become saturated in the tributary watersheds before it entered the main river channel. This saturation indicates ($SI > 0$) indicates that the dissolution of K-mica likely stops in the middle Yellow River, though neof ormation of amorphous/poorly crystalline illite could still occur, depending on kinetic pathways, microenvironments, and nucleation (Table R1).

2. K-feldspar remains undersaturated, consistent with its slow dissolution kinetics, supporting that feldspar could continue to be dissolved and to release K into river water. The stability field of K-feldspar requires lower Si ($\log a_{H_4SiO_4} < -4.5$ to -4.0 , where a refers to activity) and higher K/H. River waters typically have much Si from weathering, pushing them out of the stability field for K-feldspar and into muscovite or kaolinite stability, driving K-feldspar dissolution

(SI < 0)^{1, 2}. The lower solubility of muscovite (log Ksp ≈ -13 for dissolution) compared to K-feldspar (log Ksp ≈ -22 for microcline, adjusted for conditions) means its stabilization at higher Si levels. Feldspar dissolves to release K⁺, Al³⁺ and H₄SiO₄, pushing waters towards the muscovite stability field. However, the SI > 0 for muscovite reflects thermodynamic stability, not precipitation, as nucleation is slow^{1, 2}.

3. Illite is undersaturated and is common in freshwaters, consistent with the neoformation of only more amorphous/poorly crystalline aluminosilicate phases, rather than equilibrium with stable clay minerals, due to the variable composition and slower precipitation kinetics of illite. Although river water shows SI < 0 for illite (i.e., no precipitation), microenvironment (e.g., pore waters with higher K⁺ or lower fluid flow) can allow incorporation of K into newly-forming clays without thermodynamic saturation. As a result, illite neoformation could occur in niche environments, thereby fractionating K isotopes.

To summarize, these SI patterns arise from silica and K⁺ activities placing river waters in the muscovite stability field (SI > 0, halting dissolution) but outside the K-feldspar fields (SI < 0, enabling dissolution). Undersaturation (SI < 0) for illite stems from its lower K requirement and non-ideal structure, preventing equilibrium in rivers. These calculations support weathering models where primary silicates dissolve while secondary clays may form kinetically³ (Table R1), behaving like the simultaneous removal of Si and K which is attributed to aluminosilicate neoformation (Fig. S11). It means that the interpretation that aluminosilicate neoformation removes K remains unaffected. We revised the manuscript to better strengthen this part. RML 280-295.

Table R1 Details of K-bearing minerals. Note that all formulae are presented in ideal end-members, whereas real minerals are usually non-stoichiometric (variable K, Al, Si, Fe, Mg, H₂O, etc.).

Minerals	Ideal formula	Key structural notes
Muscovite (white mica)	KAl ₂ (AlSi ₃ O ₁₀)(OH) ₂	2:1 dioctahedral; 1 K per formula unit; Al in both tetrahedral and octahedral sites.

Minerals	Ideal formula	Key structural notes
Biotite (dark K-mica)	$K(\text{Mg,Fe})_3(\text{AlSi}_3\text{O}_{10})(\text{OH,F})_2$	2:1 trioctahedral; Mg/Fe >> Al in octahedral sheet.
Phengite (Mg-Fe-rich muscovite)	$K(\text{Al,Mg,Fe})_2(\text{Si,Al})_4\text{O}_{10}(\text{OH})_2$	Tschermak substitution: $(\text{Mg,Fe})^{2+} + \text{Si}^{4+} \rightleftharpoons \text{Al}^{3+} + \text{Al}^{3+}$
K-feldspar (Microcline / Orthoclase)	KAlSi_3O_8	Tectosilicate (framework); 1 K per 8 O; Al:Si = 1:3; no OH.
Illite (clay-size K-mica)	$\text{K}_{0.6-0.9}\text{Al}_2(\text{Si}_{3.1-3.5}\text{Al}_{0.5-0.9}\text{O}_{10})(\text{OH})_2$	Deficient K (0.6–0.9 p.f.u.); higher layer charge from tetrahedral Al; often interlayer H_2O .

3. The application of K isotopes in non-biological oceanic authigenic carbonate records. This sounds like a promising future research direction. However, it might be difficult to find a continuous record of non-biogenic authigenic carbonate in the oceans, as most carbonate is biogenic. As such, the authors may need to acknowledge this limitation, or propose a specific geological setting where such a record may exist.

Re: Accepted. As also suggested by the reviewer#3, the authigenic clay appears to be of great interest for providing such weathering archives⁴⁻⁶. We revised the manuscript following your suggestion and acknowledged its limitation. RML 372-378.

Other Comments

4. In the reply to Question 6 of Reviewer 2, the authors argued that “in low weathering-intensity regimes (W/D close to 0)... whereas the dissolved K isotopes have to be lighter than rock values.” However, in the observations (Fig. 3), the K isotope ratio is quite high at very low W/D (Fig. 3). This contrast needs to be reconciled in the revised text.

Re: Yes, the reviewer makes a key point here. Our data were analyzed in 2020, but we were hampered in understanding this observation until Li et al. (2021a,b) presented data on K isotope fractionation during dissolution and adsorption^{5, 7} and Ji et al. (2024)⁸ provide the deep groundwater circulation. Although their

experiments showed that dissolution releases light K isotope (^{39}K) during the initial dissolution stage, within several dozen hours the dissolution reaches equilibrium and there is no K isotope fractionation⁷. As we stated in your comment #2, due to the high solubility of K, a large amount of K has been incongruently dissolved into the fluids in the watersheds before arriving in the main channel. We think that the dissolved $\delta^{41}\text{K}$ at the initial period of rock-fluid interaction (hours-days) must be low (lower than the BSE/UCC), as verified in laboratory experiments⁷ but not yet in field studies. However, such incongruent dissolution of minerals creating local supersaturation of low K_{sp} elements (e.g. Fe^{3+} , Al^{3+} , and Si, thereby supplying nucleation and facilitates clay neoformation, which remove ^{39}K and leaves ^{41}K in rivers.

Therefore, the riverine $\delta^{41}\text{K}$ values are a function of two competing processes between dissolution and incorporation: In the middle stages of the rock-fluid interaction, when the river water is already saturated with respect to K-mica (Table S4)), the SPM appears to provide sufficient nucleation for kinetically-controlled aluminosilicate neoformation (Fig. 4) in microenvironments³, thus making the riverine $\delta^{41}\text{K}$ values high. On the contrary, when the SPM is low, there would be a lack of nucleation for aluminosilicate neoformation and the dissolution of secondary minerals would release ^{39}K into rivers, thus making the riverine $\delta^{41}\text{K}$ values low, and approaching to the BSE/UCC values (new Fig. 5).

5. Related to comment 4: Just out of curiosity, why could the release of light K during the initial weathering stage not be observed in the global K isotope dataset?

Re: As discussed above, we expect that the K isotopes released within the initial period of rock-fluid interaction (hours-days) is low, and lower than the BSE/UCC values (laboratory verified⁷, field verification warranted). However, based on the laboratory experiment, we predict that the release of light K during the initial weathering stage would likely only be should observable in the sheet flow or

/overland flow on the hillslope. The global dataset is based on samples collected from large rivers, which would reflect mainly by a series of transportation-sedimentation processes of weathering integrated over larger spatial and temporal scales, such that the release of light K during the initial weathering stage should not be expected to be observable in the global K isotope dataset.

6. Line 286: “the ab initial calculations³⁶.” What does “ab” mean?

Re: It is Latin, meaning “from the beginning” or “from first principles.” Hence, *ab initio* calculations refer to computations based on fundamental physical laws (usually quantum mechanics) without empirical parameters⁹. Please note that “initial” was a typo of “*initio*”. RML 254.

7. In the Methods, it is mentioned that 60 river samples were collected. However, there might not be 60 data points in Fig. 3. It may be better to present the exact number of river K isotope data in the figure caption, which may also be a requirement by the journal.

Re: Accepted. It is mentioned in Table S1 that a few samples are not available due to samples being exhausted during previous measurements. The reported data is from 51 samples, and we added this information to the new Fig. 5 caption. See Fig. 5 caption.

8. For a better understanding, the arrows in Fig. R6 of the rebuttal letter should also be indicated in Fig. S9 of the SI.

Re: Accepted. The Fig. R6 becomes Fig. S9. See the supplementary information & RML 279.

Reviewer #3 (Remarks to the Author)

I am delighted to provide some comments on the manuscript NCOMMS-25-24937A. Gou et al. reported the first seasonal riverine K isotopes, a

comprehensive and timely investigation of potassium (K) isotope systematics in the Yellow River, with the aim of exploring how seasonal variations in K isotopes can trace changes in silicate weathering intensity. The study represents an innovative and valuable contribution to the rapidly growing field of nontraditional stable isotope geochemistry and chemical weathering. On the whole, this is a novel and well-written manuscript that stands out for its originality, clarity, and scientific rigor. The authors have successfully integrated field observations, isotopic measurements, and geochemical modeling to solve the two current standing problems that, first, whether there is seasonal K isotope variation since all reported spatial data neglecting seasonal variation; and the second, data from the Yangtze River suggest K isotopes as a tracer of silicate weathering intensity, while global data don't. This research addresses the above two important knowledge gap. They observed clear seasonal K isotopes variation showing that there is huge seasonal riverine K isotopes variation, and they found how properly apply K isotopes as a tracer of weathering intensity, and does so through a carefully designed and well-executed case study on an ideal setting. Though time-financial-consuming via staying in one site for collecting those samples a whole year, such method of decreasing-variable is reasonable and thought-provoking. The discussion effectively connects riverine geochemistry to broader Earth surface processes, making the manuscript not only relevant to isotope geochemists but also to a wide community interested in weathering, erosion, and Earth system feedbacks. In their response to the reviewers' comments, the authors have provided thorough and satisfactory answers to all the questions raised and have made substantial improvements to the manuscript. I would like to see that such a study represents a significant advancement and is suitable for publication in Nature Communications.

Below are a couple of minor concerns that I encourage the authors to consider revising.

L 48, stabilize, consider revising.

Re: Accepted. RML 48.

L55-66, the study is among the first to investigate the seasonal variability of K isotopes in a large river system, while K isotopes have recently been explored as tracers of silicate weathering, most prior work has focused on global averages or on spatial catchments. By contrast, this paper presents a temporal dataset covering different seasons, hydrological conditions, and tributaries. This temporal dimension is crucial because silicate weathering intensity fluctuates seasonally with changes in temperature, runoff, and biological activity. However, the authors failed to emphasize the efforts of SEASONAL variation on controlling K isotopes fractionation mechanism. I saw your statement of weak biological control in the following paragraph, it should also be mentioned in the introduction. Further, it is recommended to slightly expand on how K isotope behavior differs from that of other alkali metals such as Li or Na, since comparisons to Li isotopes are frequently made in weathering studies. A brief clarification would further highlight why K isotopes provide unique insights.

Re: Accepted. We expanded the Introduction section to stress that previous studies mainly focused on spatial datasets, whereas our study provides a temporal perspective that captures seasonal hydrological and climatic changes. Additionally, we moved the discussion of the weak biological control, previously mentioned only in the Discussion section, into the Introduction to demonstrate the rationale for examining seasonal variability. RML 73-83.

We also agree that the comparison with Li isotopes offers useful context. However, for conciseness and to maintain a focused narrative in the Introduction, we chose not to include Li isotopes there. Instead, Li isotopes were discussed in detail in the Discussion section, where we compared the contrasting behaviors of Li and K isotopes during dissolution, transport, and secondary mineral formation. This placement allows us to fully address their differences without interrupting the main scope and the smooth flow of the

Introduction section.

L 67-76, The Yellow River sounds an ideal system for this study, but the authors seem failed to present the whole picture of the current state. For example, because of its large size, diverse lithology, strong seasonal hydrological variability, and weak human influence. Insights derived from this system are thus relevant to understanding modern continental weathering processes and their impact on global elemental cycles. I would encourage the authors add more explanation. Such as, by establishing the link between K isotopes and weathering intensity, current work provides a foundation for interpreting ancient K isotope records in marine sediments, potentially offering a new proxy for reconstructing paleo-weathering and paleo-climate conditions.

Re: Accepted. Following the suggestion, we strengthened the unique loess input, strong seasonal hydrological variability, and relatively limited anthropogenic regulation within the middle Yellow River catchment, highlighting why it is an ideal natural setting for studying weathering processes. We also now clarified that, by linking seasonal K isotope variations to changes in weathering intensity, our study provides insight into interpreting K isotope records in marine sediments, offering potential for reconstructing paleo-weathering and paleo-climate conditions. RML 67-73.

L 100, characterized, consider revising.

Re: Revised. RML 108.

L176, ~95.0% ? attention the significant figures.

Re: Revised. RML 183.

L 188-190, the K budget in the atmospheric reservoir appears mostly sourced from the biomass burning and traffic emission (Qu et al., 2024) since K is critical for organisms, I am curiosity if there is also the signal of biomass burning or

traffic emission in rivers, I acknowledge that the authors have calculated the atmospheric input to riverine reservoir as less than 1%, which may not be critical to affect the conclusion of the current manuscript, but still interesting to open other door of biological K cycles in hydrosphere.

Re: Accepted. We agree that biomass burning and traffic emissions can be significant sources of atmospheric K. However, in this study, the contribution of atmospheric K to the riverine K reservoir is quantitatively minor. As shown in our mass balance calculations, atmospheric inputs represent <1% of the total K flux to the middle Yellow River and therefore have a negligible effect on riverine $\delta^{41}\text{K}$ and do not alter our discussion and conclusion.

Nevertheless, we appreciate the reviewer's suggestion that this may open a new perspective on biological and anthropogenic K cycling within the hydrosphere. In the revision, we have added a brief statement to acknowledge that, they may be relevant in other systems with higher atmospheric deposition, though atmospheric K inputs (e.g., biomass burning, traffic emissions) are not significant in this basin, and thus merit further investigation. RML 201-207.

L 200, fertilizers, consider revising.

Re: Accepted. RML 208.

L 202-204, the traditional opinion holds that biologically light K is favored, however, more evidences show that there are both light and heavy K isotopes favored by difference vegetation leaves (Qu et al., 2022), please revise. Please explicitly mention the potential role of biological cycling (plant uptake and litter return) in modifying $\delta^{41}\text{K}$, even if it is likely minor in this context.

Re: Accepted. We recognize that both light and heavy K isotopes may be fractionated differently among plant species and tissues¹⁰. As a response, we have revised the revision to reflect this updated understanding. We now explicitly mentioned the potential role of biological cycling, including plant uptake and litter return, in modifying riverine $\delta^{41}\text{K}$ values. Our results indicate

that biological effects are minor in this semi-arid basin due to sparse vegetation, but we acknowledge that biological K cycling in other catchments merits further investigation. We have also included a figure from our previous response to reviewers (Figure R1 in the first response) as the new Figure S7 for improving clarity and presentation. RML 208-225.

L239, normalized, consider revising.

Re: Accepted. RML 247.

L 259-270, the final statement about Li–K decoupling (Fig. S9) is valuable, but there is lacking an explanation, please specify whether Li–K decoupling implies distinct controlling processes or differing sensitivities to diffusion, adsorption, or temperature, because both are proposed as a silicate weathering tracer (Dellinger et al., 2015).

Re: Accepted. As discussed above, we have revised the manuscript to clarify that the Li-K decoupling reflects fundamentally different controlling processes. Li isotopes mainly respond to dissolution and clay formation and are sensitive to diffusion due to their large relative mass difference¹¹. In contrast, K isotopes are strongly influenced by secondary mineral neoformation because K is a major element¹². Thus, although both isotopic systems are used as silicate weathering tracers, they record different aspects of weathering processes. RML 273-279 & 326-330.

L 276, behavior, consider revising.

Re: Revised. RML 286.

L 360-368, The authors end by outlining the future potential of K isotopes as both modern and paleoenvironmental proxies, an exciting and appropriate direction for continued research. Broader implication for the interpretation of K isotopes in geological archives. The results suggest that $\delta^{41}\text{K}$ variations in

marine carbonates could record temporal changes in continental weathering intensity, potentially linked to climate variability. The authors appropriately note that this interpretation must account for riverine flux weighting and secondary processes during sediment transport, but the potential of K isotopes as a paleoweathering proxy is compelling. By quantifying the link between hydrology and isotopic composition, this paper offers an important step toward integrating K isotopes into global weathering–climate feedback models. However, the authors should keep in mind that there are rare non-biological carbonates. I would suggest that the authentic clay could be served as the archivers before diagenesis, since the river water (seawater providing that the hydrothermal input keeps table, Zheng et al., 2022).

Re: Accepted. As also noted by Reviewer #2, we agree that using $\delta^{41}\text{K}$ variations in marine carbonates to record continental weathering intensity may potentially be influenced by biological processes. Following the reviewer's suggestion, we now emphasize that oceanic clay minerals likely serve as more reliable archives of $\delta^{41}\text{K}$ that may be less sensitive to diagenetic alteration, consistent with Zheng et al. (2022)¹³. Nevertheless, interpretations must still account for riverine flux weighting and secondary processes during transport and diagenesis. RML 373-378.

The final sentence is quite long and dense. Splitting it into two sentences would improve readability, one emphasizing the constancy of the fractionation factor, and another highlighting its application for reconstructing paleo-seawater $\delta^{41}\text{K}$ and weathering feedbacks

Re: Accepted and revised. RML 369-378.

L 360, conceptualized, considering revising.

Re: Accepted. RML 369.

Acknowledgments: We are grateful to two anonymous reviewers and editor

Rebecca Neely, and helps from Dr. Xin-Yuan Zheng, and Shilei Li during the revision, for their valuable constructive suggestions that substantially strengthened our manuscript.

References mentioned in the replies above:

1. Stefánsson A, Arnórsson S. Feldspar saturation state in natural waters. *Geochimica et Cosmochimica Acta* **64**, 2567-2584 (2000).
2. Kampman N, Bickle M, Becker J, Assayag N, Chapman H. Feldspar dissolution kinetics and Gibbs free energy dependence in a CO₂-enriched groundwater system, Green River, Utah. *Earth and Planetary Science Letters* **284**, 473-488 (2009).
3. Lu M, Diao Q-F, Zheng Y-Y, Yin Z-Y, Dai Z. Influence of water on tensile behavior of illite through the molecular dynamics method. *International Journal of Geomechanics* **24**, 04024024 (2024).
4. Santiago Ramos DP, Nielsen SG, Coogan LA, Scheuermann PP, Seyfried WE, Higgins JA. The effect of high-temperature alteration of oceanic crust on the potassium isotopic composition of seawater. *Geochimica et Cosmochimica Acta* **339**, 1-11 (2022).
5. Li W, Liu X-M, Hu Y, Teng F-Z, Hu Y. Potassium isotopic fractionation during clay adsorption. *Geochimica et Cosmochimica Acta* **304**, 160-177 (2021).
6. Li W, Liu X-M, Wang K, Fodrie FJ, Yoshimura T, Hu Y-F. Potassium phases and isotopic composition in modern marine biogenic carbonates. *Geochimica et Cosmochimica Acta* **304**, 364-380 (2021).
7. Li W, Liu X-M, Wang K, Koefoed P. Lithium and potassium isotope fractionation during silicate rock dissolution: An experimental approach. *Chemical Geology* **568**, 120142 (2021).
8. Ji T-T, Jiang X-W, Han G, Li X, Wan L, Wang Z-Z, Guo H, Jin Z. Contrasting behavior of K isotopes in modern and fossil groundwater: Implications for K cycle and subsurface weathering. *Earth and Planetary Science Letters* **626**, 118526 (2024).
9. Zeng H, *et al.* *Ab initio* calculation of equilibrium isotopic fractionations of potassium and rubidium in minerals and water. *ACS Earth and Space Chemistry* **3**, 2601-2612 (2019).
10. Qu R, Han G. Potassium isotopes in herbaceous plants: A potential new tool for C₃

and C₄ plant research. *Journal of Geophysical Research: Biogeosciences* **127**, e2021JG006682 (2022).

11. Bourg IC, Richter FM, Christensen JN, Sposito G. Isotopic mass dependence of metal cation diffusion coefficients in liquid water. *Geochimica et Cosmochimica Acta* **74**, 2249-2256 (2010).
12. Li W, Liu X-M, Hu Y, Teng F-Z, Hu Y-F, Chadwick OA. Potassium isotopic fractionation in a humid and an arid soil–plant system in Hawai'i. *Geoderma* **400**, 115219 (2021).
13. Zheng X-Y, Beard BL, Neuman M, Fahnstock MF, Bryce JG, Johnson CM. Stable potassium (K) isotope characteristics at mid-ocean ridge hydrothermal vents and its implications for the global K cycle. *Earth and Planetary Science Letters* **593**, 117653 (2022).